# Cryo-EM structure of the human Kv3.1 channel reveals gating control by the cytoplasmic T1 domain

Gamma Chi[1,2], Qiansheng Liang[3], Akshay Sridhar[4], John B. Cowgill [4], Kasim Sader[5], Mazdak Radjainia[5], Pu Qian[5], Pablo Castro-Hartmann[5], Shayla Venkaya[1,2,6], Nanki Kaur Singh[1,2], Gavin McKinley[1,2], Alejandra Fernandez-Cid[1,2,7], Shubhashish M. M. Mukhopadhyay[1,2,6], Nicola A. Burgess-Brown[1,2,7], Lucie Delemotte [4], Manuel Covarrubias[3] & Katharina L. Dürr [1,2,8✉]

Kv3 channels have distinctive gating kinetics tailored for rapid repolarization in fast-spiking neurons. Malfunction of this process due to genetic variants in the *KCNC1* gene causes severe epileptic disorders, yet the structural determinants for the unusual gating properties remain elusive. Here, we present cryo-electron microscopy structures of the human Kv3.1a channel, revealing a unique arrangement of the cytoplasmic tetramerization domain T1 which facilitates interactions with C-terminal axonal targeting motif and key components of the gating machinery. Additional interactions between S1/S2 linker and turret domain strengthen the interface between voltage sensor and pore domain. Supported by molecular dynamics simulations, electrophysiological and mutational analyses, we identify several residues in the S4/S5 linker which influence the gating kinetics and an electrostatic interaction between acidic residues in α6 of T1 and R449 in the pore-flanking S6T helices. These findings provide insights into gating control and disease mechanisms and may guide strategies for the design of pharmaceutical drugs targeting Kv3 channels.

[1] Centre for Medicines Discovery, Nuffield Department of Medicine, University of Oxford, Roosevelt Drive, Oxford OX3 7DQ, UK. [2] Structural Genomics Consortium, Nuffield Department of Medicine, University of Oxford, Roosevelt Drive, Oxford OX3 7DQ, UK. [3] Department of Neuroscience and Vickie and Jack Farber Institute for Neuroscience, Sidney Kimmel Medical College at Thomas Jefferson University, Philadelphia, PA 19107, UK. [4] Department of Applied Physics, Science for Life Laboratory, KTH, Solna, Sweden. [5] Materials and Structural Analysis, Thermo Fisher Scientific, Achtseweg Noord 5, 5651 GG Eindhoven, Netherlands. [6] Present address: Exscientia Ltd., The Schrödinger Building, Heatley Road, The Oxford Science Park, Oxford OX4 4GE, UK. [7] Present address: Exact Sciences Ltd., The Sherard Building, Edmund Halley Road, The Oxford Science Park, Oxford OX4 4DQ, UK. [8] Present address: OMass Therapeutics, Ltd., The Schrödinger Building, Heatley Road, The Oxford Science Park, Oxford OX4 4GE, UK. ✉email: katharina.duerr@omass.com

High frequency firing of action potentials in the central nervous systems is essential for a multitude of physiological functions, ranging from early development and axonal path finding in neuronal progenitors[1], learning and memory in hippocampal neurons[2], sensory processing in auditory nuclei[3] and specialized cells of the retina[4], to circadian rhythms in the suprachiasmatic nucleus of the hypothalamus[5,6]. The precise interplay of ligand-gated ion channels and a diversity of voltage-gated ion channels with different activation thresholds is required to tailor the frequency of action potential firing to each of these physiological processes. Voltage-gated $K^+$ channels of the Kv3 family represent an important subfamily among the 70 different Kv channel genes in the human genome, with specialized biophysical properties to sustain high-frequency firing of action potentials[7,8]. The combination of ultra-fast activation and deactivation kinetics and high activation potential of these channels are key for their ability to repolarize the membrane and terminate the action potential. Even slight changes in the biophysical properties of Kv3 channels can affect their ability to sustain high-frequency firing in different brain regions and can causes severe neurological disorders, including ataxias, epilepsies and schizophrenia[9–15].

Like other pore-forming subunits of Kv channels, Kv3 members are comprised of an N-terminal cytoplasmic T1 domain, which is involved but not required for $Zn^{2+}$-mediated tetramerization[16,17], followed by the voltage-sensing domain (VSD) with transmembrane helices S1–S4 which is connected via a helical linker (S4/S5L) to the pore forming domain (PD) of the channel. The latter contains the selectivity filter (SF) for potassium ions at the end of the pore helix (PH) and two transmembrane helices S5 and S6. The lower S6 segment has a conserved PXP motif ("PVP" in Kv3 channels) which forms a kink and provides flexibility for essential gating movements in this region[18–21].

Due to the overall similarity in architecture with other Kv channels, it has been unclear which structural determinants are responsible for the unique biophysical properties of Kv3 channels. Nevertheless, a number of functional studies have addressed this question by analysing chimeric constructs with channels from different Kv subfamilies or other voltage-dependent proteins. For example, a study by Labro et al. [22] showed that Kv3.1 channels exhibit ultrafast voltage-sensor relaxation, resulting in resurgent $K^+$ currents during repolarization and this property was linked to the S3 and S4 loop of the voltage sensor domain. Another study showed that the fast activation and deactivation kinetics could be transplanted onto the voltage-sensing phosphatase Ci-VSP if the S3/S4 portion of the enzyme was replaced by the respective Kv3.1 region[23]. This is not surprising considering that the length and composition of the S3/S4 linker has been identified as major factor shaping characteristics of the VSD in the Shaker $K^+$ channel[24]. However, some of the largest sequence-differences to other members of the Kv1–4 families (Fig. S1) are within the cytoplasmic T1 domain, the C-terminal extension beyond S6, and the so-called turret domain, which is located between S5 and the PH helix. In the longer splice variant of Kv3.1 channels (Kv3.1b), an axonal targeting motif (ATM), which is located C-terminal of S6, was shown to interact with ankyrin G and promote enrichment of the channel in the axon[25]. The shorter Kv3.1a variant however is retained in somatodentritic membranes, resulting in lower frequency firing[26]. It was demonstrated that this is caused by a masking of the ATM motif in Kv3.1a through interactions with the N-terminal T1 domain[25]. Interactions between N- and C-terminal domains have been suggested for other Kv channels[27], but this region is unresolved in the currently available T1-containing Kv1 channel structures[19,28–30]. For the Kv1 subfamily of potassium channels, characterization of T1-deleted Shaker variants[31] has ruled out an essential role of the T1 domain for normal gating beyond the well-studied role of the N-terminus for N-type inactivation[32]. However, multiple studies for other Kv subfamilies have provided experimental evidence suggesting that the cytoplasmic domains may play a major role in controlling channel gating[33–38], but structural data corroborating this hypothesis is currently missing.

Here, we present cryo-EM structures of the human Kv3.1a channel in an activated state, providing mechanistic insights into the modulation of Kv3 channels by the cytoplasmic T1 domain. The structure reveals multiple interactions of the most C-terminally located helix in T1 (α6) with the S4/S5 linker, as well as with residues within the S6 extension (S6T) containing the axonal targeting motif. The structure also shows the architecture of the turret domain in Kv3.1, featuring a unique interface with the extracellular S1/S2 linker of the VSD. Together with functional studies of aptly chosen mutants and molecular dynamics simulations, these findings illuminate the electromechanical coupling mechanism in Kv3.1 channels and provide a molecular explanation for the characteristic gating properties of Kv3 channels.

## Results

**Overall structure and cytoplasmic T1/ATM interaction.** To elucidate the structural determinants underlying the biophysical properties of Kv3 channels, we purified the shorter α-isoform of Kv3.1 from mammalian cells and determined cryo-EM structures under different sample conditions (apo, $Zn^{2+}$-containing vs. $Zn^{2+}$-free, see Supplemental Table 1). We achieved 3D reconstructions between 3.1 and 3.5 Å for the different conditions (Figs. S2 and S3), which show only minor differences in the overall structure, and no clear densities for additional $Zn^{2+}$ sites other than the known sites in the T1 domain[16] (see Supplementary discussion). In the following structural analysis, we thus focus mostly on observations from the best resolved structure (Fig. 1a) which was obtained in digitonin and in the presence of 400 μM $Zn^{2+}$ (representative densities for different regions of the channel are shown in Fig. S4). The locations of Kv3.1a residues affected by disease mutations in the *KCNC1* gene are illustrated in Fig. 2d and in more detail in Fig. S5.

Kv3.1a exhibits a typical domain-swapped Kv channel architecture which brings the VSD from one subunit of the channel close to the S5/S6 helices of the PD from an adjacent subunit (Fig. 1a, Movie M1), similar to the structures of rat Kv1.2[19] and the Kv1.2-Kv2.1 chimera, in which the voltage-sensing S3b/S4 segment from Kv2.1 has been grafted onto Kv1.2 for structure determination[29]. However, a striking difference to all previously published T1-containing full-length Kv structures[19,28,39] is the relative orientation between TMD and the T1 domain. When superposing the TM region of Kv3.1a with the respective region in the structure of the Kv1.2-Kv2.1 chimera (Fig. 1b), the T1 domain of Kv3.1a is rotated clockwise by ~48° with respect to the T1 domain of Kv1.2-2.1 (Fig. 1c). Superposition of the isolated T1 domains highlights the different orientation of the α6 helices of the two channels (Fig. 1d), whereas the rest of the domain exhibits a similar fold in both structures (RMSD = 0.7). Small conformational differences to Kv1.2-2.1 and Kv1.3 are also present in α3 of T1, and the different conformation seen in Kv3.1a would result in a clash with the β-subunits present in the Kv1.2-2.1 and Kv1.3 complex structures (Fig. S6a). As a result of these differences, the α6 helix in the context of the full Kv3.1a channel is rotated by an almost 90° angle compared to Kv1.2-2.1, bringing it in much closer proximity to the S4/S5 linker (Fig. 1b).

Due to this rotation of T1 in Kv3.1a, the extension beyond S6 that contains the ATM motif is located near α1, α4 and the α5/α6 linker of T1 from a neighbouring subunit (Fig. 1a, e, f, Fig. S7).

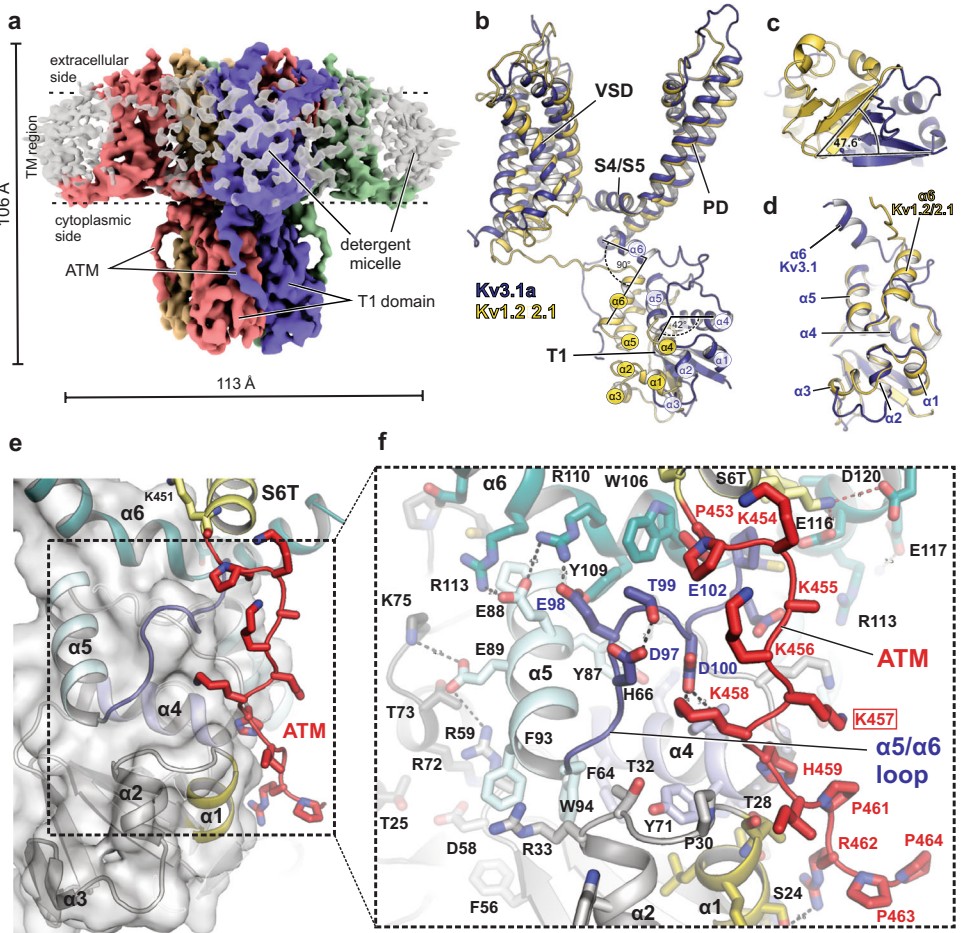

**Fig. 1 Kv3.1a full-length structure showing alternative T1 arrangement compared to Kv1.2/2.1 and electrostatic ATM/T1 interactions. a** Density maps of the human Kv3.1a channel, coloured by chain. TM region is illustrated by dashed lines. **b** Superposition of a single chain of Kv3.1a (dark blue cartoon) with one protomer of the Kv1.2-2.1 paddle chimera (pdb: 6EBK, shown in yellow cartoon representation), aligning the S4/S5 linker and pore domain. **c** Cytoplasmic view of the T1 domain of Kv3.1a superposed on the Kv1.2/2.1 structure (yellow cartoon) using the same alignment as in (**b**). Angle indicates rotational displacement measured between a vector defined by cα atoms of G16/H22 in Kv3.1 and G41/Q47 in Kv1.2-2.1, respectively. **d** Superposition of an individual T1 protomer (residues 8–120, dark blue) from Kv3.1 with the respective region of the Kv1.2-2.1 protomer (yellow). **e** Detailed view of the ATM/T1 contact region. T1 is shown as semi-transparent surface representation/cartoon and the ATM motif C-terminal from S6T is shown as red cartoon. **f** Close-up view of the inset from **e**, illustrating electrostatic contacts between acidic residues in α5 of the N-terminal T1 domain and a cluster of lysines (K454–K458) within the ATM of the C-terminus. Red frame indicates location of frameshift variant K457fs associated with EPM7 (ClinVar ID: 692088).

A cluster of acidic charged residues (D97, E98, D100, E102) in the α5/α6 linker of T1 creates a negatively charged patch on T1 which electrostatically attracts the positive charges from a series of five lysines in the ATM (K454–K458). K458 forms a salt-bridge with D100 and further C-terminal the ATM/T1 interaction is stabilized by a polar contact between R462 and S24 in α1 (Fig. 1f). In a subclass of particles, we were able to tentatively trace more of the C-terminus for two diagonally related chains of the tetramer (Fig. S4h, i), showing that this extension is stabilized by a series of intrasubunit interactions with residues in α2 and α3 of the T1 domain (Fig. S7). Despite the overall close prediction of the Kv3.1a structure by AlphaFold2[40], the confidence of the model is low for the ATM region, and the interactions with T1 seen in our structure were not faithfully modelled (Fig. S8a, b). This region is also unresolved by the EM maps of Kv3.1a reported in a recent preprint[41].

Of note, T1/C-terminal interactions have been suggested for other Kv channels, including Kv2.1[42] and Kv4.1[27]. While the current work was under review, structures of full-length human Kv4.2 were published[43], confirming the close proximity of T1 and the C-terminal end of S6T, which are connected via an inter-subunit

salt bridge located near α6 of Kv4.2 (Fig. S6c, d). The T1 domain of Kv3.1 shows electrostatic surface properties distinct from Kv1 family members for the central vestibule flanked by α4 helices in T1 (Figs. S9 and S10a–c). Kv4.2 and Kv4.3 share more similar electrostatic properties to Kv3.1, with acidic residues in the α4/α5 loop contributing to the overall negative surface potential (Fig. S10a, d, e), whereas the same region in Kv1 structures is dominated by positive surface charges (Figs. S9d, f and S10b, c). In contrast T1 domains from Kv1 and Kv4 channels which have characteristic salt bridges to stabilize α4 helices from neighbouring subunits against each other (Fig. S10b–e), this interaction is missing in Kv3 channels due to substitutions by uncharged residues (Fig. S10a, inset).

We hypothesized that the negative electrostatic potential arising from a ring of D81 residues in the Kv3.1 lower vestibule (Fig. 2c, inset) may play a role in preventing local $K^+$ depletion after repetitive firing of fast-spiking action potentials[44] and could help maintain the large conductance characteristic for Kv3 channels[45]. In most other Kv channels, this residue is replaced by neutral amino acids, i.e. Asn (Kv1.2/Kv1.3), Gln (Kv4.1), His (Kv4.2) or Tyr (Kv4.3), and only Kv2.1 has also an acidic residue (Glu) in the corresponding position (Figures S1, S10). To

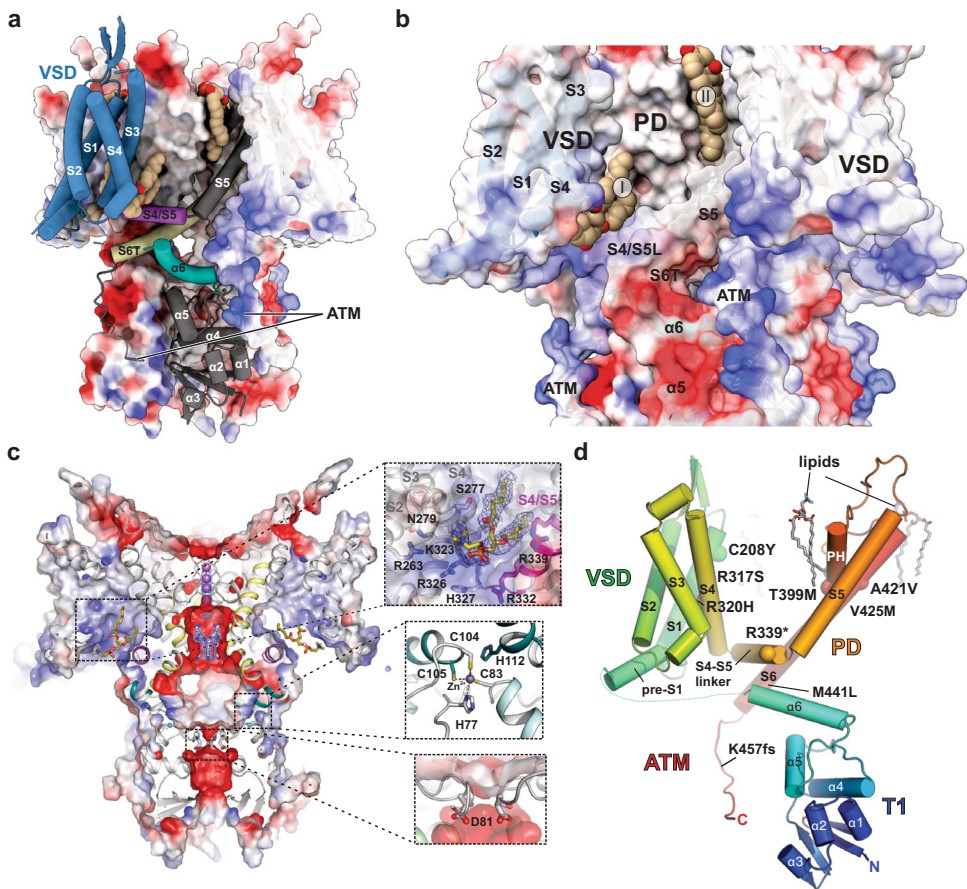

**Fig. 2 Electrostatic surface properties and location of disease mutations. a** Electrostatic surface representation of the Kv3.1a tetramer with a single protomer shown as cartoon, highlighting the relative arrangement of VSD (blue), S4/S5 linker (magenta) and α6 (teal) of the T1 domain. Lipids are shown in brown space-fill representation. The surface is coloured by electrostatic potential (red, −5 kT e$^{-1}$; blue, +5 kT e$^{-1}$). **b** Enlarged view of the Kv3.1 tetramer in full surface representation, highlighting electrostatic attraction between negatively charged α5 and α6 in T1 (red surface areas) and positively charged ATM (blue surface areas). Two lipid-binding sites are shown: site I between S4 and S4/S5 linker and site II at the interface between VSD and PD. **c** Slab view of (**a**) with insets 1. highlighting residues involved in lipid interactions at site I, 2. the interfacial Zn$^{2+}$ binding site in T1 and 3. the lower constriction in T1 with D81 as major player for negative surface potential and local K$^+$ ion abundance. **d** Kv3.1a protomer in cartoon representation with cα atoms of residues implicated in disease mutations shown as spheres. Lipids are shown in stick representation.

investigate the functional importance of D81, we performed MD simulations of K$^+$ ions accessing the pore region from the cytoplasm via the T1 fenestrations. The results from a simulation with wild-type Kv3.1a show an abundance of K$^+$ ions in the lower vestibule (Fig. S11d), whilst the introduction of a charge-inverting mutation (D81K) causes a significant drop of the of K$^+$ occupancy at the interstitial fenestrations (Fig. S11e). Furthermore, TEVC recordings with oocytes injected with equal amounts of mRNA show decreased ionic currents at full activation for this mutant, compared to the wild-type (Fig. S11f–h). As predicted by the MD simulations, this result is consistent with a decrease in unitary conductance to be confirmed by analysis of single channel currents. We propose that D81 may play a role as K$^+$ reservoir to sustain repolarizing currents in fast-spiking neurons.

**Voltage sensor domain conformation and pore arrangement in Kv3.1a.** Consistent with the high activation potential of Kv3.1, the S4 helix of the VSD is in a less activated conformation compared to Kv1.2/Kv2.1 and Kv4.2, with only three (R1–R3) of the four voltage-sensing charges located above the conserved F256 in S2 (Figs. 3a, b and S12g), which is part of the hydrophobic charge transfer centre (CTC) defined in previous Kv channel structures[46]. In Kv3.1, the positively charged side chain R4 (R320) is engaged in a cation–π interaction with the aromatic

ring of F256 and forms an electrostatic interaction with D282 (located in S2, see Fig. S12d). The corresponding R4 side chain (R303) in Kv1.2/2.1 is located above the conserved Phe and interacts with E226 (in S1, Fig. S12b). A similar interaction is observed for Kv3.1a between residues E249 and R3 (R317). As a result, the S4 sensor in Kv3.1 is shifted downwards by one 3$_{10}$ helical turn compared to the more activated Kv1.2-2.1 channel structure (Fig. S12c, f). Of note, gating charge mutations R317S and R320H are both loss-of-function variants linked to human channelopathies (Figs. 2d and S5)[10,11,47]. Another notable difference is that a second glutamate, E213 in the S2 segment of Kv3.1a, replaces T184 in the Kv1.2 structure (Fig. S12a, b). It may therefore neutralize positive gating charges in S4 when the VSD assumes a different activation state than the one captured in the current structure, in which the residues are too far apart for this putative interaction. Interestingly, both S2/S3 and S3/S4 linkers in Kv3.1 are shorter and exhibit a more structured and compact conformation compared to Kv1.2/2.1, in particular S2/S3 which is mostly helical, similar to the conformation in the Na$_v$Ab channel[48]. These areas also show substantial sequence differences and therefore explain how their transplantation on Kv channels of other families can confer gating properties like those of Kv3.1a. Another interesting difference in our Kv3.1a structures is that the backbone carbonyl oxygen of gating charge R6 (R326) interacts

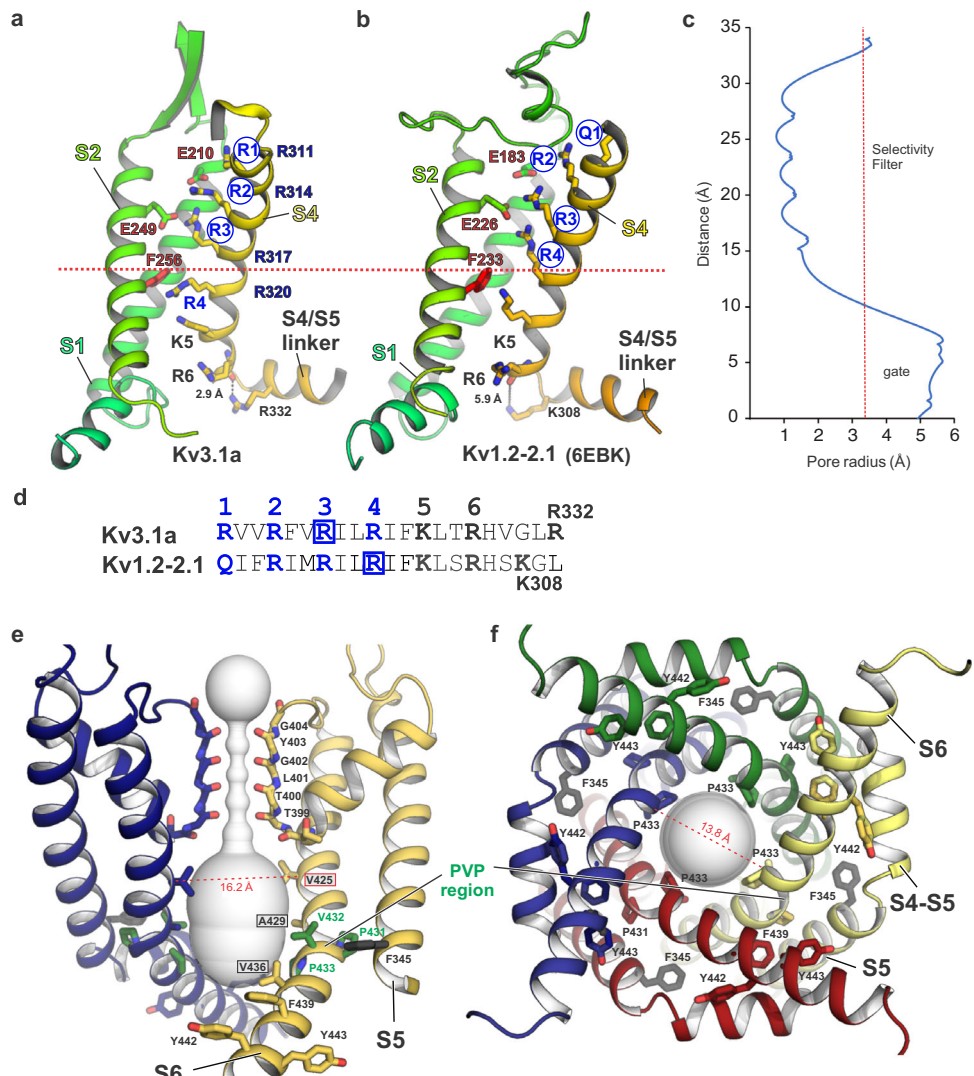

**Fig. 3 Voltage sensor domain arrangement and pore properties of Kv3.1a. a** Cartoon representation of the VSD S1–S4 (S3 is omitted for clarity) in Kv31a, showing the relative positioning of the gating charges R1–R4 (blue labelled) in S4 (shown in yellow) with respect to conserved residues (F256 and E249) of the hydrophobic charge transfer centre (CTC), located in S2 (shown in light green). Blue circles highlight voltage-sensing gating residues located above the position of conserved F256 of the CTC, which is represented by the dotted red line. **b** Cartoon representation of the VSD arrangement in Kv1.2-2.1 (pdb 6EBK), highlighting that all four voltage-sensing charges R1–R4 (blue circles) are located above F233 compared to only three (R1–R3) in Kv3.1a shown in panel (**a**). **c** Plot of the Kv3.1a pore radius, as determined by MOLE[77]. Red dotted line marks the radius of hydrated K[+]. **d** Sequence of S4 residues for Kv3.1a (top) and Kv1.2-2.1 (bottom) with the main voltage-sensing R/Q residues within the conserved arginine-rich motif highlighted in blue. Blue boxes encircle voltage-sensing arginines which are located above the CTC. **e** Cross-membrane view of the Kv3.1a pore domain and selectivity filter. For clarity, only two chains are shown. The pore diameter is shown as light grey surface representation. Residues in the PVP motif are shown as green sticks, major pore lining residues are highlighted by black frames. Conserved aromatic residues in the lower S6T gate are shown as sticks. F345 in S5 (unique to Kv3 family) is shown as black stick representation. Dashed line indicates cα distance between V425 (framed in red), a residue relevant for the gain-of-function mutation V425M associated with epileptic encephalopathy[15]. **f** Cytoplasmic view of the open pore with pore-lining S6 helices from all four subunits shown as cartoon representation. Dashed line indicates cα distance between P433 from two diagonal subunits.

with the positively charged side chain of R332 in the S4/S5 linker. This interaction is absent in Kv1.2-2.1 because the respective S4/S5 linker sequence lacks an arginine in this position and the nearby K308 side chain is too far away for a similar interaction (Fig. 3a, b, d). In the Kv4 subfamily, however, an analogous stabilizing interaction could exist because the R332 residue in the S4/S5 linker is conserved. This is confirmed by the recent full-length Kv4.2 structure[43] which indeed shows a salt bridge between the corresponding R6 (R305) carbonyl oxygen and the side chain of R311 in the S4/S5 linker (Fig. S12g).

Despite the less activated S4 position in the VSDs, the lower S6 gate of Kv3.1 is captured in a dilated conformation, exhibiting a

pore radius near the PVP region large enough to accommodate a hydrated K[+] ion (Fig. 3c, e, f). MD simulations confirm that the pore is found in a stable conductive state, displaying a sustained hydration of the inner gate over MD simulation timescales (Fig. S11a–c). Our maps also show densities in the selectivity filter which likely correspond to K[+] ions from the purification buffer (Supplementary Fig. S10a). Opening of the lower S6 gate however is unexpected, because of the high activation potential of Kv3.1a ($V_{0.5} = +8\,mV$, see Fig. 6d), and consequently only a small fraction of the channels (20–25%) should be activated at 0 mV. Since we observe distinct densities below the selectivity filter, it is possible that the lower gate of Kv3.1a is propped open by an

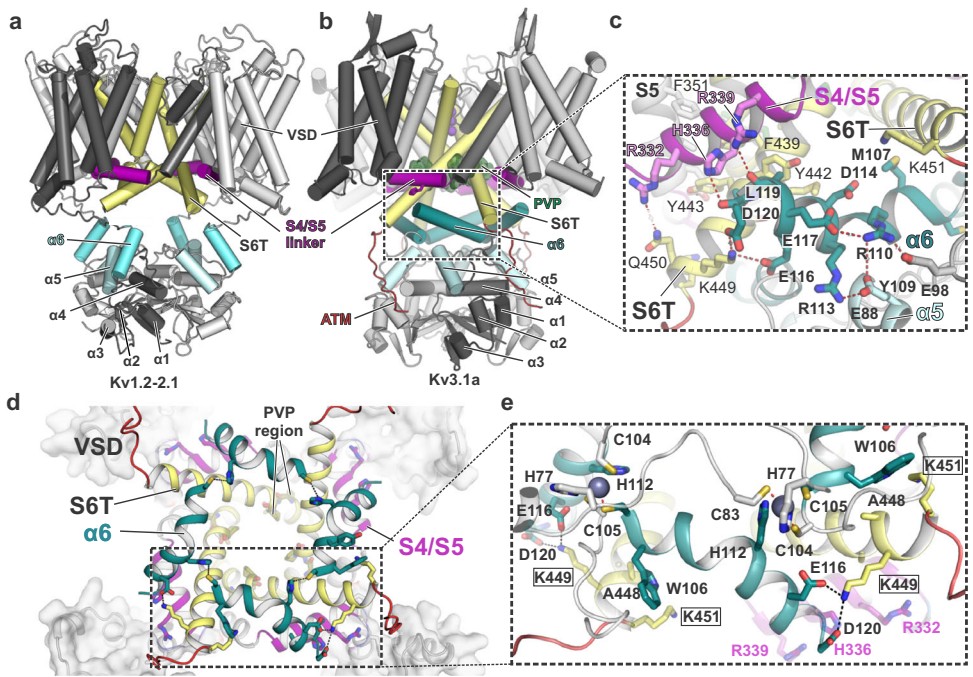

**Fig. 4 The α6 helix of the Kv3.1 T1 domain forms a second cuff below the S4/S5 linker, stabilized by inter- and intrasubunit interactions. a** Cartoon representation of the Kv1.2-2.1 tetramer (light grey) with one protomer highlighted in dark grey. S4/S5 linker is represented in magenta, S6 is coloured in pale yellow and α6 of the T1 domain is shown in aquamarine blue. **b** Cartoon representation of the Kv3.1a tetramer, illustrating the unique orientation of α6 (teal blue) of the T1 domain. Colour-coding of other elements is the same as in (**a**). **c** Inset from **b** with an enlarged view of the α6 helix and its interactions with residues in the S4/S5 linker (magenta) and the lower gate formed by S6T (pale yellow). **d** Cytoplasmic view of the tetrameric "α6 gating cuff" (shown as teal cartoon), located below the gating machinery formed by S4/S5 (magenta) and S6T (yellow). Only helix α6 of T1 is shown for clarity. **e** Inset from **d**, detailing inter-subunit interactions between α6 helices from adjacent subunits and interactions between α6 and S6T. Zn²⁺ ions are shown as grey spheres.

unidentified molecule trapped within the pore (see Fig. S10a and Supplementary Results). Similar densities are also present in the open pores of the Kv1.2/2.1 and Kv4.2 structures determined by cryo-EM[28,43].

Interestingly, two disease mutations are located between SF and the PVP motif (A421V[12] and V425M[15], see Fig. 2d). V425M is a gain-of-function mutation with leftward-shifted activation potential (by −30 mV) and higher sensitivity towards inhibition by the open pore blocker fluoxetine[15]. Since the side chain of V425 is one of the key pore lining residues (Fig. 3e), a replacement by the larger methionine could introduce steric clashes when the lower S6 gate is closed, hence reducing the threshold for opening and explaining the gain-of-function phenotype of the mutation, as well as the higher potency of fluoxetine in blocking the mutant channels.

**S4/S5 linker and lower S6 gate interact with α6 of the T1 domain.** The unique T1 arrangement in Kv3.1a brings α6 near two areas of the TM domain with known importance for gating and electromechanical coupling, i.e., the C-terminal end of S6 (S6T) and the S4/S5 linker. In contrast to Kv1.2/2.1 where α6 of the T1 domain is located distant from these regions (Fig. 4a), the α6 helices in Kv3.1a create a ring-like structure which runs right below the cuff formed by the S4/S5 linker helices (Fig. 4b, d). In addition to the zinc-coordinating interactions of C104/C105 in α6 and H77 in the α4/α5 linker (Fig. 2c, middle inset and Fig. 4e), two acidic residues (E116 and D120) at the C-terminal end of α6 are forming salt bridges to K449 in S6T (Fig. 4c, e). The side chains of W106 and M107 located at the other end of α6 are engaged in hydrophobic interactions with A448 in S6T of a neighbouring subunit (Fig. 4e). Due to this arrangement, the

interactions mediated by either end of the four α6 helices in T1 may stabilize the lower S6 gate in the open state.

Sequence alignments with other Kv channels reveal that three methionines in S6 are unique and conserved within the Kv3 subfamily (M430, M441, M447 in Kv3.1a, see Fig. 5a–d). M430 near the PVP region has been identified as a major factor for pore stability in Kv3.1a, because mutation to leucine caused flicker of the currents[49]. Our structure may provide a molecular explanation for this phenotype, because the partial charge on the side chain sulfur in M430 forms a sulfo-aromatic interaction with the edge of the aromatic ring from F345 in S5 (Fig. 5c). This interaction exists only in Kv3 channels, because both residues are leucines in other Kv channels and it may strengthen the interlock between S5, S6 and S4/S5 linker. Whilst M430 is a known factor for open pore stability, the roles of methionines M441 and M447 in S6T is less clear and merits further investigation in future studies, because M441L is a known variant associated with EPM7 but currently of unknown functional significance (ClinVar ID 848761).

Notably, the three-way interaction between α6, S6T, and S4/S5 linker is only possible due to the proximity of these helices, which in turn is facilitated by an abundance of alanine residues within this region: A446 and A448 in S6T, A115 and A118 in α6, and A340 in the S4/S5 linker (Figs. 4e and 5). The compact side chains in these hydrophobic residues allow a tight packing of α6 against Y442 in S6T. The two alanines in α6 and S6T are unique in Kv3 channels (Fig. 5d and alignment in Fig. S1). Interestingly, the C-terminal end of α6 is also rich in acidic residues creating a negatively charged surface area which interacts with the polar side of the amphipathic S4/S5 linker. The latter carries excess positive charges from the side chains of R332 and R339 and H336 (Figs. 4c and S9A). Furthermore, exposed backbone carbonyls of

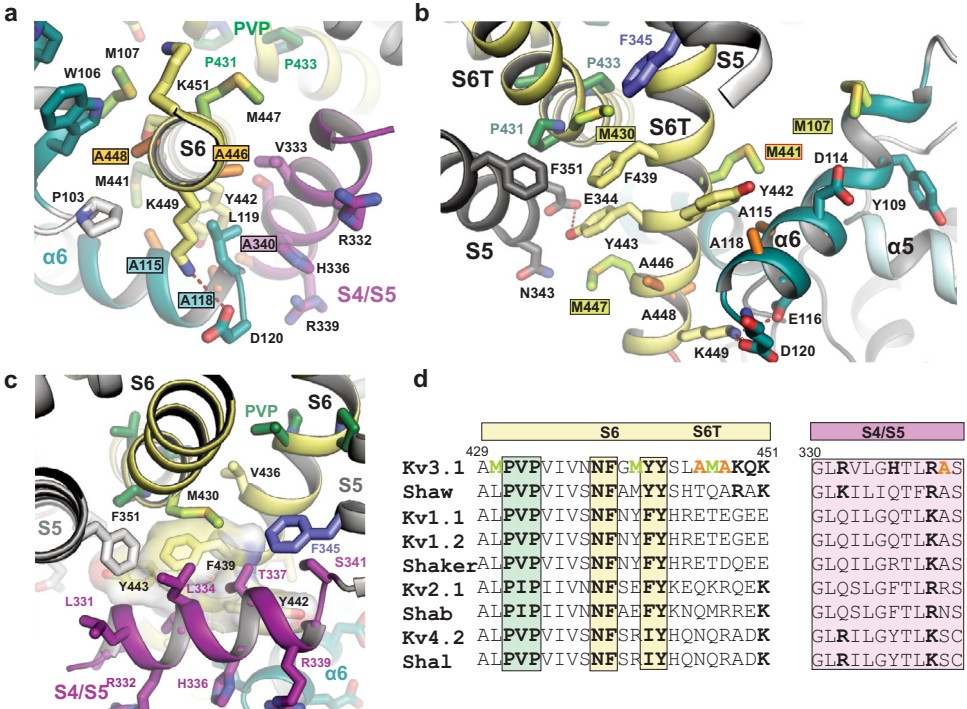

**Fig. 5 A cluster of Kv3-specific alanines allows three-way interactions between α6, S4/S5 linker and S6T and location of Kv3.1-specific methionines in S6T. a** Cytoplasmic view onto S6T (pale yellow cartoon) of the lower gate in Kv3.1a, highlighting the close proximity to neighbouring helices S4/S5 and α6 of the T1 domain, which is possible due to the compact side chains of A115 and A118 (α6), A340 (S4/S5 linker) and A446 and A448 (S6T). The close contact allows the formation of a stabilizing salt bridge between D120 in α6 and K449 in S6T. **b** Rotated view from **a**, highlighting the abundance of Kv3-specific methionines in S6T (M430, M441, M447) and α6 (M107). S4/S5 linker is omitted for clarity. Red frame indicates the location variant M441L associated with EPM7. **c** Illustration of the hydrophobic surface created by aromatic S6T residues F439, Y442, and Y443 (displayed as semi-transparent surfaces) which cradles the S4/S5 linker (magenta) close to the PVP motif (green sticks) and the α6 helix in the T1 domain. F345 in S5 (slate blue) and M430 (lime green) from adjacent subunits form a sulfo-aromatic interaction. **d** Sequence alignment of the S6/S6T (left) and S4/S5 linker (right) regions of Kv3.1a with other T1-containing Kv channels (human or fruit fly). Kv3-specific alanines and methionines are highlighted in orange and lime, respectively. Conserved residues across other Kv families are shown in bold, including the PVP motif (green) and the aromatic S6T residues involved in electromechanical coupling interactions with the S4/S5 linker.

L119 and D120 interact with the imidazole ring of H336 in the centre of the S4/S5 linker (Fig. 4c). The exposure of these backbone carbonyls is a result of the partial unwinding at the C-terminal end of α6. In our maps, the density beyond D120 is weak, indicating structural flexibility. The recent Alphafold2 model predicts a slightly longer helical region, albeit with low confidence (Fig. S8b, d). This matches well with our MD simulations showing that this region can be helical as well as without secondary structure, with the latter enabling backbone interactions with S4/S5L and S6T.

**Role of inter-domain interactions for open state stabilization and fast activation/deactivation.** To test the functional significance of these interdomain interactions for Kv3.1a voltage-dependent gating, we characterized a series of alanine substitutions by TEVC in *Xenopus* oocytes (Figs. 6 and S13, S14). In terms of voltage dependence, the most profound effects are caused by replacements of R332 in the S4/S5 linker and K449 in S6T. The clear shift of the respective G/V curves of these mutants to more positive potentials (Fig. 6a, b) suggests that the interaction between R332 and the backbone of S4 voltage sensor R6 (contact A in Fig. 7b), as well as the S6T/α6 contact mediated by K449 (contact B in Fig. 7b) could be important for efficient coupling to the S4 movement in the VSD. For both mutants, we observe significantly slower activation time constants (Fig. S14e, g), consistent with the depolarizing shift in $V_{0.5}$. For R332A, we also observe faster

deactivation time constants which is consistent with the even more pronounced depolarizing shift in the activation curve compared to K449A for which the effect on deactivation kinetics is small. To investigate the significance of the contact B further, we analysed MD simulations at depolarizing and hyperpolarizing potentials. Upon a 4.5 μs application of depolarizing voltages, we observe upward movement of R1–R4 in two out of four subunits of Kv3.1a (Fig. S15a and Movie M2). The same subunits also exhibited a significantly higher propensity of the E116/K449 salt bridge (Fig. 7b), indicating that the contact gets stronger with S4 activation. Under deactivating conditions, we observe that the propensity of the interaction is lower when the gating charges in S4 move downward relative to the CTC (Figs. S15b, c and 7c, d), suggesting that there is a correlation between S4 activation and the presence of the E116/K449 salt bridge.

Furthermore, we examined the effect of mutations in the respective α6 residues involved in contact B and observed small changes in $V_{0.5}$ for E116A, D120A, and the E116A/E120A double mutant (Fig. 6c, d). Whilst the individual mutations E116A and D120A have only mild effects (Fig. S14b, d), the E116A/E120A double mutant shows a substantial change in the activation and deactivation kinetics across the whole voltage range (right panel in Fig. 6f), indicating that the combined charge-neutralization at these α6 residues affects the gating kinetics of Kv3.1a.

Notably, we observed a similar "sluggish" gating phenotype for mutant H336A located at the centre of the S4/S5 linker (Fig. 6e).

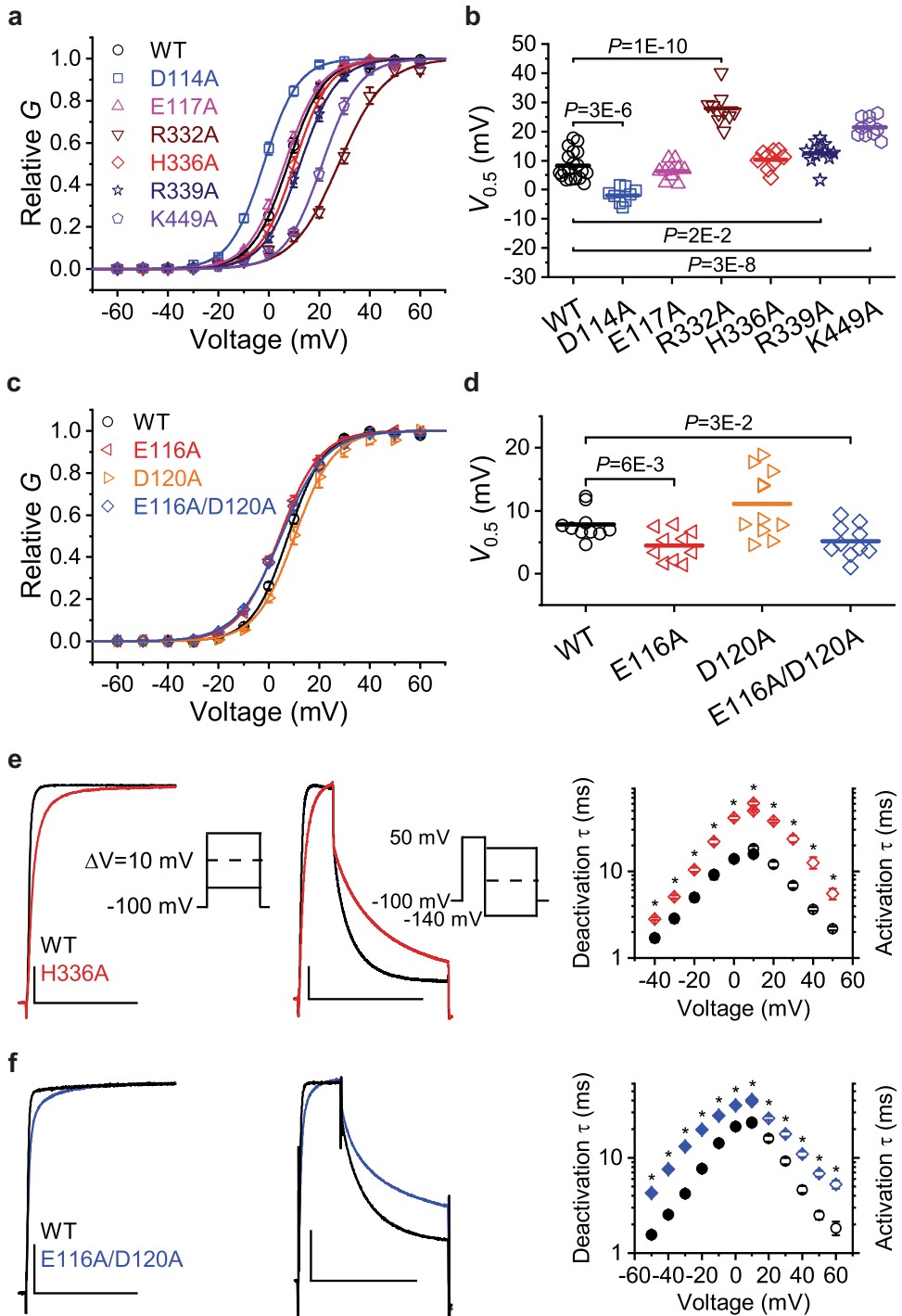

The mutation is situated at a mechanistically important position and interrupts polar interactions with several backbone carbonyl oxygens in the α6 helix (Fig. 4c). In the AlphaFold2 model, H336 interacts with the side chain and the backbone carbonyl of S121 at the C-terminal end of the slightly elongated α6 helix predicted by the algorithm (Fig. S8b, d). This might be a consequence of the different S4/S5 linker conformation in the Alphafold2 model, which captures S4 in a more activated state (Fig. S8c). Hence, H336 may change its interaction partners in α6 to accommodate structural rearrangements in S4 and the attached S4/S5 linker which occur in response to a change in the membrane potential.

Interestingly, mutant R339A at the C-terminal end of S4/S5L does not affect the voltage-dependence of activation substantially (Fig. 6a, b), but it shows significantly faster activation and deactivation kinetics (Fig. S14f). The phenotype suggests that the mutation lowers the energy barrier for activation and deactivation. The alanine replacement approach also identified a charge-neutralizing mutant in α6 (D114A) which causes a substantial hyperpolarizing shift in the activation potential (Fig. 6a, b), faster activation time constants and slower deactivation time constants (Fig. S14a), indicating a relative stabilization of the open state.

**Turret and S1/S2 linker create a second VSD/PD interface for efficient electromechanical coupling.** A recently described disease mutation of Kv3.1[12] is located at the C-terminal end of S1, close to the S1–S2 loop C208Y (Figs. 2d and 8d). This cysteine is

**Fig. 6 Mutational and electrophysiological analyses of key residues involved in functional inter-domain interactions between α6, S4/S5 linker and S6T. a** Normalized $G–V$ curves ($G/G_{max}$) of Kv3.1a WT and the indicated mutants. The symbols and error bars represent mean ± SEM ($n = 9-19$ oocytes), and the solid lines are the best fits assuming a first-order Boltzmann equation (see the "Methods" section). **b** Scatter graphs comparing of $V_{0.5}$ values from WT and several Ala mutants. Each symbol represents a measurement from an individual oocyte, and the overlaid horizontal line represents the mean value. In each group, $n = 9-19$ oocytes. $P$ values are indicated on the graph. **c** Normalized $G–V$ curves for the indicated α6 mutations as described for panel (**a**). In each group, $n = 10-11$ oocytes. **d** Scatter graph for the indicated α6 mutations as described for panel (**b**). In each group, $n = 10-11$ oocytes. **e** Left: Overlay of representative whole-oocyte currents from Kv3.1a WT and the mutant H336A at 50 mV. These currents were evoked by the voltage protocol shown in the inset. Centre: Overlay of representative whole-oocyte currents from Kv3.1a WT and the mutant H336A while the repolarizing voltage is 0 mV. From a holding voltage of −100 mV, these currents were evoked by a 30-ms step depolarization to +50 mV, which was immediately followed by a 100-ms repolarizing step. To determine the voltage dependence of deactivation, the repolarizing step was changed in 10-mV increments from −140 to +30 mV (inset). The horizontal and vertical scale bars indicate 100 ms and 1 μA, respectively. Right: The voltage dependence of the time constants of activation (hollow symbols) and deactivation (filled symbols) from Kv3.1a WT and H336A. **f** Left and centre: overlay of representative whole-oocyte currents from Kv3.1a WT and the double mutant E116A/D120A. Protocol details as described for panel (**e**). The horizontal and vertical scale bars indicate 100 ms and 1 μA, respectively. Right: The voltage dependence of the time constants of activation (hollow symbols) and deactivation (filled symbols) from Kv3.1a WT and E116A/D120A. Symbols and error bars in panels e and f represent the mean ± SEM and * indicates $p < 0.001$. Source data are provided as a Source Data file. Exact $n$ and $p$ values for panels **a**–**d** are listed in Supplemental Table 2.

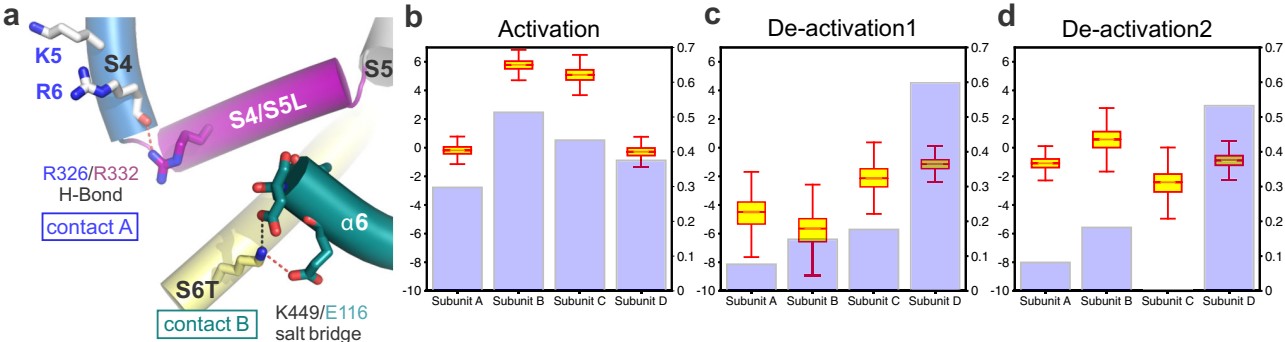

**Fig. 7 MD simulations of S4 voltage sensor movement during activation/deactivation reveal a correlation with E116/K449 salt bridge stability. a** Cartoon representation of the S4–S5/S6T contact region in Kv3.1a, showing the positions of the R326/R332 H-bond and the salt bridge E116A/K449 between α6 in the T1 domain and S6T. **b** Comparison of the Z-position of the R4 dating charge relative to the CTC (boxplot) and the stability of the E116/K440 (violet bars) for each subunit under activating conditions (+300 mV). Data calculated from 750 ns (12,000 frames) following the movement of R4 Arginine (SI Fig. S15). The salt-bridge was assumed to form in frames with distance between the heavy atoms of the residues being <3.3 Å. Boxes span the interquartile range and whiskers extend up to 1.5 times this range. **c**, **d** Comparison of the Z-position of the R4 gating charge relative to the CTC (boxplot) and the stability of the E116/K440 (violet bars) for each subunit under deactivating conditions (−650 mV) from two independent trajectories. Data calculated from 750 ns (12,000 frames) following the movement of R4 arginine. The salt-bridge was assumed to form in frames with distance between the heavy atoms of the residues being <3.3 Å. Boxes span the interquartile range and whiskers extend up to 1.5 times this range. Source data for **b**–**d** are provided as a Source Data file.

conserved in other Kv channels and was suggested to be part of a second, co-evolved interface between VSD and pore domain, with a major role in electromechanical coupling[50]. In Kv3.1, C208 has been identified as a key residue involved in inhibition by extracellular Zn$^{2+}$[51]. In our EM structure, C208 is indeed located at interacting distance to a conserved proline located of the pore helix (Fig. 8a, d). Kv3 channels have a conserved histidine (H212 in Kv3.1a) and glutamate (E213 in Kv3.1a) in this region, which are contributing additional polar interactions to strengthen the VSD–PD interface further (Fig. 8d). Interestingly, mutant H212A results in a dramatically increased activation constant and reduced current amplitude[51], thus hinting at a prominent role of this interface for ultra-fast activation of Kv3 channels. Another set of Kv3-specifc histidines is located nearby, in the S5-PH linker region (H381 and H383, see Fig. 8b). Again, mutation of these histidines to alanines was shown to affect activation kinetics and the H381A/H383A double mutant had the same dramatic phenotype as the H212A mutant[51]. The side chains of H381 and H383 are interacting with Q372 and Q409, respectively, and H383 also forms a H-bond with Y407, which in turn interacts with Y403 near the K selectivity filter (Fig. 8a, b).

This region is also known as the turret domain of Kv channels and only members of the Kv3 subfamily contain an extra stretch of amino acids which is absent in other Kv families (Fig. 8c, e). A serine uniquely present in Kv3.1 (S377) and Kv3.2 interacts with E238 in the S1/S2 β-hairpin (Fig. 8a). It must be noted that the S1/S2 loop is flexible and therefore resolved at lower resolution than most of the channel (see local resolution maps in Figs. S2, S3, 2, 3). 3D variability analysis was utilized to identify a small subset of particles with a 20° rotated S1/S2 linker region compared to the rest of the particles. In this alternative conformation, S377 forms a hydrogen bond with the backbone carbonyl of R237 (Fig. 8f). None of the other published Kv channel structures show the same S1/S2 β-hairpin structure, indicating that the more structured conformation of this loop in Kv3 channels could be of functional importance for the turret interactions and may explain why this is a hot spot for a range of disease variants (Fig. S5).

**Lipid interactions and potential drug binding sites in Kv3.1a.** In all our Kv3.1 EM maps we observe clear densities for a native lipid in the pocket between PH and VSD (Site II in Fig. 2, and Fig. S4a). The polar head group of the lipid is stabilized by interactions with the imidazole group of H212 in the S1/S2 linker and backbone interactions with Q409 and M414 in S6 (Fig. 9a). The non-polar side chains of W411 in S6 and W392, P388, and

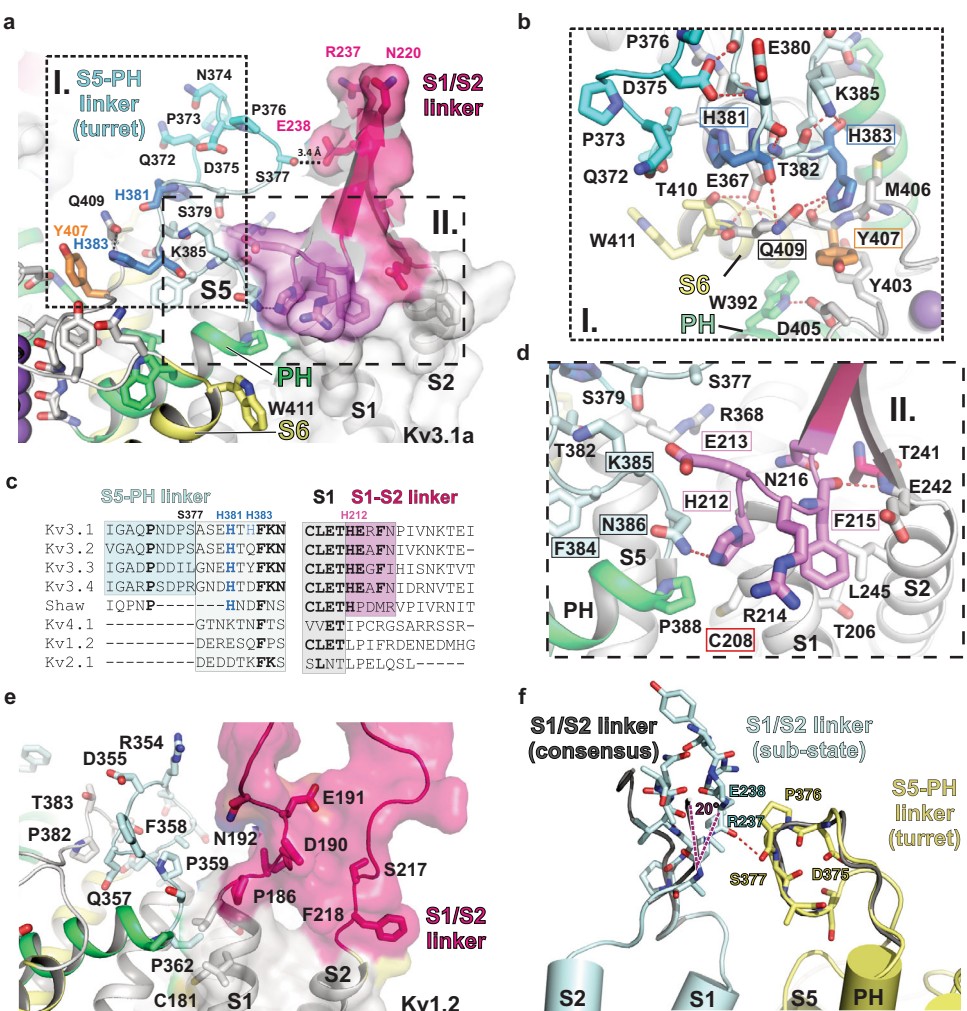

**Fig. 8 Architecture of the turret domain of Kv3.1a and its interaction with the S1/S2 linker of the adjacent VSD. a** Overview of the S5-PH linker region ("turret region", shown in shades of blue) and the interface between pore helix PH (green cartoon) and S1/S2 linker of the VSD (surface representation and pink cartoon). **b** Inset I from **a**: Detailed view of histidines H381 and H383 (bold blue sticks) in the turret domain and their interactions with Q409 in S6 (pale yellow) and Y407 (orange sticks) near the selectivity filter of the pore domain. **c** Sequence alignments of the S5-PH linker (left) and the S1/S2 linker region (right) for human Kv3 family members and other human T1-containing Kv channels, highlighting the enlarged turret domain unique to the Kv3 subfamily and the location of conserved Kv3-specific histidines. Conserved residues are represented in bold letters. Coloured regions follow colour-coding in panels **a**, **b**, **d**. **d** Inset II from **a**: detailed view of the PD/VSD interface formed by residues at the N-terminal end of the S1/S2 linker of the VSD unique to Kv3 subfamily (pink ribbon) and their interactions with residues in the transition zone between the turret domain (light cyan) and the pore helix (PH in green). Red frame indicates the location of variant C208Y associated with non-progressive myoclonus epilepsy[12]. Pink frames and blue boxes highlight conserved residues in Kv3 channels. **e** View of the shorter S5-pore helix linker region of Kv1.2 ("turret region", shown in pale blue) and the interface between pore helix PH (green cartoon) and S1/S2 loop of the VSD (surface representation and cartoon in bold pink). **f** Alternative conformation of the S1/S2 linker (light blue cartoon representation) from a subset of particles obtained by 3D variability analysis in comparison to the consensus conformation (dark grey cartoon representation). The sub-state allows the formation of an H-Bond (red dashed line) to the backbone carbonyl of S377 in the turret domain (represented as yellow cartoon). Purple dashed lines illustrate the angle between the main conformation (consensus) and the sub-state of the S1/S2 linker.

F391 in PH are interacting with the hydrophobic tails of the lipid (Fig. 9a). Interestingly, the site is analogous to the anionic lipid interaction site capable of modulating the prokaryotic analogue KcsA[52] and to the PC binding site in the Kv1.2/Kv2.1 structure[53] (Fig. S16b). Furthermore, it strongly resembles the binding pocket determined for a TRPC6 agonist[54] which mimics the action of the native lipid agonist DAG (Fig. 9b). Analogous small molecule binding sites near the S6/PH intersubunit interface have also been identified for TRPML1 (agonist)[55] and CavAb (antagonist)[56]. This highlights the functional importance of this region in other ion channels and could hint at a druggable pocket near the turret domain. A recent preprint with the structure of Kv3.1a in

complex with a positive modulator[41] suggests that the molecule is bound near the S4/S5 linker, close to a second site containing non-protein densities in our structure (Figs. S4b, S16a, Site I in Fig. 2b). The density in our maps can likely be attributed to a native lipid with its head group in close proximity to R326, corresponding to R6 of the conserved arginines in S4 helix (Fig. 2c, top inset). It is also within 5 Å of R332 in S4/S5 linker helix, which we identified as a key residue in controlling activation potential. We carried out coarse-grained MD simulations in a complex lipid bilayer approximating the lipid composition of membranes in eukaryotic organisms, to study which lipids might bind in the sites revealed by cryo-EM. Whereas anionic lipids

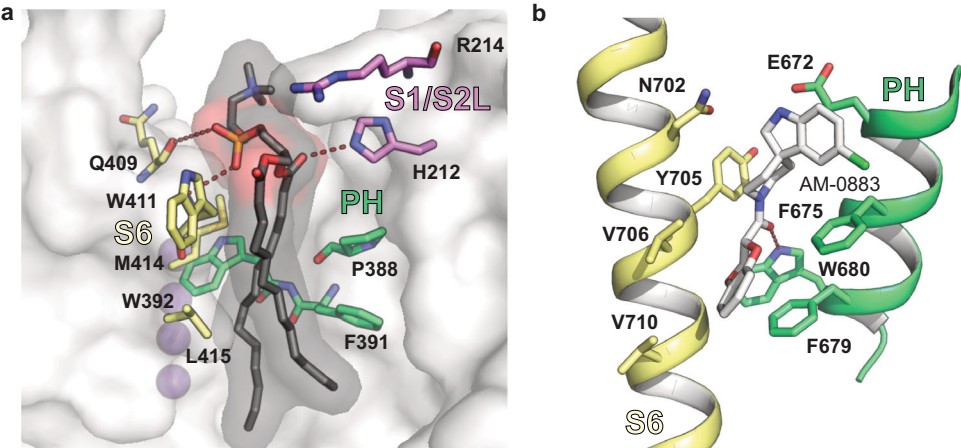

**Fig. 9 Lipid binding site near the turret domain of Kv3.1a resembles known drug binding sites in other channels. a** Kv3.1a residues in the pore domain (S6 in yellow, PH in green) involved in hydrophobic interactions with the fatty acid tails (black) of the lipid bound to the groove at the PD/VSD interface. Residues from the nearby S1/S2 linker are coloured in pink. $K^+$ ions are represented as purple spheres. **b** Binding site for agonist AM-0883 (white) between S6 (yellow) and PH (green) of TRPC6 (pdb: 6UZ8).

predominately interact with Kv3.1 residues at site I near the S4/S5 linker, there is no preferential localization at site II (Fig. S16c, d) —an interesting result considering the role of anionic lipids at this site within the prokaryotic channel KcsA. CG free-energy perturbation calculations further pointed towards an absence of preference for anionic lipids in site II: the estimation of the free-energy difference related to the alchemical transformation of zwitterionic PC into anionic PA yielded $-1.69 \pm 0.67$ kJ/mol. On the other hand, the differences in lipid binding characteristics in site I are in line with its similarity to the $PIP_2$ pocket in Kv7.1[57] (Fig. S16a) and may be utilized for the development of pharmaceutical drugs designed to specifically target one of the two lipid-binding pockets in Kv3.1a.

## Discussion

Kv channels are quintessential regulators for neuronal excitability, and diverse members of the family have evolved to fine-tune the threshold potential for firing, the duration of action potentials and firing rates for specific cell functions. Kv3 channels stand out against the rest of the family due to several kinetic properties which are tailored to sustain high-frequency firing (up to 1 kHz). Here, we present structural insights from cryo-EM structures of the human Kv3.1 structure and illuminate several hot spots in the channel architecture that may play important roles for: (i) the ultrafast activation and deactivation kinetics, (ii) the large conductance which aids efficient repolarizing activity, and (iii) the positive activation threshold to ensure that Kv3 channels only open during the repolarization phase of the action potential.

Interestingly, Kv3 channels exhibit only a small difference in the $V_{0.5}$ values for their gating charge–voltage ($Q/V$) and conductance–voltage ($G/V$) relationships[22], indicating tighter electromechanical coupling compared to other Kv channels in which the curves are further apart[58]. The stronger interlock between S5 and S6 through intersubunit interactions of residues F345 (S5) and M430 (S6) could be one contributor to tighter coupling between VSD and PD in Kv3 channels, since the mechanical pulling force exerted on the pore-flanking S6 helices is likely increased by this connection. The "α6 cuff" observed due to the unusual T1 arrangement in the Kv3.1a structure could certainly be another determinant for promoting this concerted opening of the pore via its clamp-like action to control the stability of S6T in the closed and open states. The cooperativity

between subunits will be enhanced because each α6 helix is in direct contact with two neighbouring pore-flanking S6 helices. It is conceivable that this cross-bridging of the helical S6 bundle could potentially accelerate the final concerted step associated with opening of the pore and could therefore provide a mechanical explanation for the fast activation kinetics of Kv3.1a. In line with this interpretation, we expect that destabilizing this cooperative arrangement by mutagenesis would likely affect both kinetics and the relative stability of open/closed states.

We indeed observe consistently altered G/V curves for several mutants disrupting the interdomain interactions between the residue K449 in S6T and multiple residues in the negatively charged α6 helix of T1, hence supporting the idea that the lower gate is stabilized in the open state as suggested by our structures. We also observe sluggish gating for the double mutant E116A/D120A in α6. This phenotype suggests that fast activation/deactivation characteristics of Kv3.1 may depend on the cooperative interactions mediated by α6 residues and that a disruption of this cooperative network might increase the energy barrier of the activation/deactivation process.

Moreover, our structural analysis and electrophysiological data indicate that the tetrameric arrangement of α6 helices in T1 may play a role in controlling Kv3.1 gating properties through interactions with the S4/S5 linker. Using alanine mutagenesis, we show that disruption of an α6 interaction facilitated by H336 in the S4/S5 linker slows both activation and deactivation kinetics of Kv3.1a. We also observe that mutation of another residue in the S4/S5 linker (R332) causes accelerated deactivation and decelerated activation which result in a profound depolarizing shift in activation potential.

These findings are ground-breaking in light of the current model of electromechanical coupling in Kv channels, where the transmembrane region is thought to be primarily responsible for the transmission of signal between the VSD and the pore: the S4/S5 linker is the moving part that couples the conformational changes in the S4 voltage sensors to the pore domain via interactions with the S6 tail region[59,60] and direct interactions between S4 and S5 have been proposed as a non-canonical coupling pathway[61–63]. Due to the remarkable degree of sequence conservation of the S4/S5 linker in Kv3 channels across species, this region was previously suggested to be linked to the specialized Kv3 properties and driving their evolution in vertebrates[8]. Our experimental data indeed confirm that the linker contains unique residues involved in functionally important interactions with an

extra control element in Kv3.1a, namely the "cuff" of α6 helices in the T1 domain. According to our structures, the latter is ideally positioned to influence conformational changes of both S4/S5L linker and S6T and this interpretation is further supported by our MD and electrophysiological data. Another region of interest for potential enhanced electromechanical coupling in Kv3 channels is the interface between turret domain and the S1/S2 β-hairpin, which could be key for anchoring the immobile S1/S2 helices in the VSD and therefore allow more efficient S4 movement in response to voltage changes.

Finally, our structures show that the unique T1 arrangement in Kv3.1a enables physical contact to the C-terminal axonal targeting motif. A frameshifting variant in the *KCNC1* gene associated with progressive myoclonic epilepsy 7 (EPM7, ClinVar ID: 692088) causes premature termination at K457 (located within the ATM, Fig. 1f) and therefore deletes many of the residues involved in this T1/ C-terminal interaction. The phenotype of this variant suggests that either the full extension is required for proper axonal targeting or to maintain normal gating properties of Kv3.1 channels, but the mutation has not yet been functionally characterized. The interface between ATM and T1 also harbours a range of single point mutations identified in patients with EPM7; with S44N, H45Y and F46L in α3 helix of T1 domain and G467R, N470S and S474Y in ATM reported to date (Fig. S5). Therefore, our structure warrants exploration of the functional significance of ATM beyond intracellular targeting. Based on the structures and biophysical analyses reported here, additional studies would be needed to fully understand the intricate gating mechanism of Kv3 channels and its implications for the ability to sustain high frequency firing.

In summary, the findings from the present work allow us to map disease mutations of in the human *KCNC1* gene and understand the underlying molecular mechanisms. Our results may also inspire future studies beyond Kv3 channels because a role of the cytoplasmic T1 domain in modulating gating was demonstrated for other Kv channels[34–36]. Since we identified previously unknown control elements in Kv3 channels, our insights could be transformative for structure-guided drug discovery targeting these novel structural features. Hence, the structures will serve as a blueprint for the rational design of new pharmaceutical drugs against a multitude of channelopathies and other severe CNS disorders linked to malfunction of high-frequency firing.

## Methods

**Molecular biology, virus production and protein expression**. Full-length human Kv3.1 (isoform A) was cloned from the mammalian gene collection (cDNA clone MGC:129855 IMAGE: 40024733) into LIC-adapted pHTBV C-terminally tagged Strep-II/10-His/GFP vector. Baculoviral DNA from the transformation of DH10Bac with the plasmid was used to transfect Sf9 cells grown in Sf-900™ II media supplemented with 2% foetal bovine serum (Thermo Fisher Scientific). The resulting virus was further amplified by transducing Sf9 cells followed by incubation on an orbital shaker at 27 °C for 70 h, followed by harvesting by centrifugation at 900×*g*.

An Expi293F™ cell culture in mid-log phase (2 × 10⁶ cells mL⁻¹) in Freestyle 293™ Expression Medium (Thermo Fisher Scientific) was infected with high-titre baculovirus (3% v/v) in the presence of 5 mM sodium butyrate. Cells were grown in orbital shaker at 37 °C with 8% CO₂ for 38 h before being harvested by centrifugation at 900×*g* for 10 min. The pelleted cells were washed with phosphate-buffered saline, pelleted again, then flash-frozen in liquid nitrogen (LN₂) for storage in −80 °C freezer.

**Purification of Kv3.1a channels**. Whole cell pellet expressing Kv3.1a was resuspended to a total volume of 50 mL per 15 g cell pellet with buffer A (20 mM HEPES pH 7.5, 100 mM NaCl, 50 mM KCl) supplemented with 0.7% w/v lauryl maltoside neopentyl glycol (LMNG; Generon) and 0.07% cholesteryl hemisuccinate (CHS; Generon) for solubilization. The cells were solubilised at 4 °C for 1 h with gentle rotation, then centrifuged at 45,000×*g* for 1 h. Washed Strep-Tactin Superflow (IBA) was added to the lysate to a ratio of 0.4 mL resin per 100 mL lysate, and the slurry was gently rotated at 4 °C for 1 h. The resin was collected on a gravity flow column and washed with buffer B (buffer A with 0.003% LMNG and 0.0003%

CHS), then with buffer B supplemented with 2 mM ATP and 5 mM MgCl₂. Kv3.1 was eluted with 6 CV of buffer B supplemented with 5 mM D-desthiobiotin followed by overnight tag-cleavage with TEV protease.

The sample was then concentrated and subjected to a size-exclusion chromatography pre-equilibrated with buffer C (buffer A supplemented with 0.04% digitonin (Apollo Scientific)). Peak fractions were pooled and concentrated to 20 μM.

**Cryo-electron microscopy sample preparation, data collection, and data processing**. For Kv3.1 sample in apo state, 20 μM sample was directly used. For EDTA-incubated sample, 100 mM EDTA was added to the 20 μM Kv3.1 sample for a final concentration of 1 mM EDTA, then it was incubated on ice overnight. For ZnCl₂-incubated sample, 40 mM ZnCl₂ was added to the 20 μM Kv3.1 sample for a final concentration of 400 μM ZnCl₂, then it was incubated on ice overnight.

All samples were frozen on Quantifoil Au R1.2/1.3 200-mesh grids freshly glow-discharged for 30 s, with plunge freezing performed on Vitrobot Mark IV (Thermo Fisher Scientific) set to 100% humidity, 4 °C, 20 s wait time and 2.5–4.5 s blotting time.

The apo-Kv3.1 dataset was collected with EPU 2 on a Titan Krios (Thermo Fisher Scientific) operating at 300 keV at Midlands Regional Cryo-EM Facility (MRCEF; Leicester, UK). 4,043 super-resolution dose-fractionated micrographs (0.42 Å pixel⁻¹) were collected on a K3 (Gatan) detector at ×105,000 nominal magnification. Micrographs were binned to 0.84 Å pixel⁻¹ in Relion 3.0.8 during motion correction with 5 by 5 patches using MotionCor2. Motion-corrected images were imported to Cryosparc v2.14.2[64], then the defocus values were determined by Patch CTF function. 1,539,126 particles were picked with blob picker function, and extracted particles were subjected to three cycles of 2D classification. 58,244 particles from good classes were used to generate an ab-initio model. The particles were then subjected to a heterogeneous refinement with two classes, where the good class contained 49,327 particles. These were then used for non-uniform refinements with C1 and C4 symmetries using the ab-initio model as reference, with final resolutions reaching 3.5 and 3.2 Å, respectively.

The EDTA-Kv3.1 dataset was collected with EPU 2 on a Titan Krios (Thermo Fisher Scientific) operating at 300 keV at Cambridge Nanoscience Centre (CNC; Cambridge, UK). 7214 super-resolution dose-fractionated micrographs (0.42 Å pixel⁻¹) were collected on a K3 (Gatan) detector at ×105,000 nominal magnification. Micrographs were motion-corrected with Patch Motion Correction on Cryosparc v3.1.0, and defocus values were estimated with Patch CTF function on Cryosparc. 3,177,434 particles were picked with blob picker function, and extracted particles were subjected to two rounds of 2D classification. 217,788 particles from good classes were used to generate an ab-initio model. The particles were subjected to non-uniform refinement with C4 symmetry using the ab-initio model as reference, which resulted in a 3.2 Å reconstruction. For the subclass with extended axonal targeting motif, the particles from refinement were symmetry-expanded by C4 symmetry, which were then used for Cryosparc's 3D variability analysis with a mask including Kv3.1's cytoplasmic domain. Cluster function was used to isolate 149,365 particles, after which remove duplicate particles function was used to obtain 110,585 unique particles. These particles were used for a non-uniform refinement with C1 symmetry using one of the maps from 3D variability analysis as reference, resulting in a 3.6 Å reconstruction. For the subclass with interaction between S1/S2 linker and turret domain, the symmetry-expanded particles were subjected to 3D variability analysis with a mask including Kv3.1's transmembrane domain. Cluster function was used to isolate 107,532 particles, after which remove duplicate particle function was used to obtain 93,461 unique particles. These particles were used for a local refinement function with the 3D variability analysis map as reference and transmembrane domain mask, which resulted in a 3.6 Å reconstruction.

The Zn-Kv3.1 dataset was collected with EPU 2 on a Titan Krios (Thermo Fisher Scientific) operating at 300 keV at Cambridge Nanoscience Centre (CNC; Cambridge, UK). 5010 super-resolution dose-fractionated micrographs (0.42 Å pixel⁻¹) were collected on a K3 (Gatan) detector at ×105,000 nominal magnification. Micrographs were motion-corrected with Patch Motion Correction on Cryosparc v3.1.0, and defocus values were estimated with Patch CTF function on Cryosparc. 2,285,988 particles were picked with blob picker function, and extracted particles were subjected to two rounds of 2D classification. 224,524 particles from good classes were used to generate an ab-initio model. In order to separate Kv3.1 monomers from dimers, the particles were reextracted to larger box size (900 × 900 pixels downsampled to 450 × 450 pixels) then heterogeneous refinement was performed. 133,488 monomer particles were used for non-uniform refinement with C4 symmetry, resulting in a 3.1 Å reconstruction. For the dimer subclass, 91,036 dimer particles were further cleaned with remove duplicate particles function, leading to 72,764 unique particles. These were used for a non-uniform refinement with C4 symmetry, resulting in a 3.1 Å reconstruction.

**Model building and refinement**. Transmembrane domain of a monomer model of Kv1.2 (PDB ID: 3LUT) and a tetramerisation domain of a Kv3.1 orthologue from *a. californica* (PDB ID: 3KVT[16]) were fitted to the apo-Kv3.1 map on UCSF Chimera[65], then combined to a single PDB file. The sequence was substituted with Kv3.1's sequence with CHAINSAW, and the model was subsequently manually built in Coot[66]. The model was refined with Phenix real space refine, and its

geometry was verified in Phenix[67] with MolProbity. This model was used as reference for EDTA-Kv3.1 and Zn-Kv3.1 maps. Models were manually refined in Coot, which were then refined with Phenix real space refine. The models' geometry were verified in Phenix (MolProbity).

**Heterologous expression in *Xenopus* oocytes and electrophysiological recordings**. The human Kv3.1a cDNA in pcDNA3.1(+) was kindly provided by Nadia Pilati (Autifony Therapeutics, Ltd.). This cDNA encodes the full-length Kv3.1a plus a C-terminal FLAG tag. The QuikChange site-directed mutagenesis kit (Agilent, Santa Clara, CA, USA) was used to create all mutations except K449A. Kv3.1a-K449A and the corresponding WT without the FLAG tag were generated synthetically in the pcDNA3.1(+) vector (GenScript, Piscataway, NJ). The voltage dependence of Kv3.1a with or without the C-terminal FLAG tag is similar (Supplementary Fig. S13a, b, i). Capped mRNA was synthesized in vitro by using mMessage mMachine kit (Ambion, Austin, TX) and then microinjected into oocytes using the Nanoject II (Drummond Scientific, Broomall, PA)[68]. Care and surgery of *Xenopus laevis* frogs was performed according to a protocol approved by the Thomas Jefferson University IACUC.

Whole-oocyte currents were recorded under two-electrode voltage-clamp (TEVC) conditions using the OC-725C amplifier (Warner Instrument, Hamden, CT, USA) 1–2 days post mRNA injection as previously reported[58]. Data acquisition, P/4 on-line leak subtraction and initial analysis were performed using pClamp 10.3 (Molecular Devices, Sunnyvale, CA, USA). The extracellular solution contained (in mM): 96 NaCl, 2 KCl, 1.8 CaCl$_2$, 1 MgCl$_2$, 5 HEPES, 2.5 sodium pyruvate, adjusted to pH 7.4 with NaOH. The electrodes were filled with 3 M KCl. All recordings were conducted at room temperature (21–23 °C).

**Data analysis for electrophysiological recordings**. Data analysis, curve plotting, and fitting were performed in Clampfit (pClamp 10.3, Molecular Devices, Sunnyvale, CA, USA) and OriginPro 9.6 (OriginLab, Northampton, MA, USA). Conductance-voltage relations (G–V) curves were characterized by the best-fit to the first-order Boltzmann equation:

$$G(V) = \frac{G_{max}}{1 + e^{\frac{(V_c - V_{0.5})}{k}}} \qquad (1)$$

where $G_{max}$ is the max conductance and $V_c$ is the command voltage. The best-fit to this equation returned the $V_{0.5}$ and $z = 25.5/k$. To obtain the time constants of current activation at various depolarizing command voltages, we obtained the best-fit to the rising phase of the current (excluding the short initial current lag) assuming a first-order exponential function or a sum of exponential terms. Typically, no more than two terms were necessary to obtain a satisfactory fit. A similar approach was used to obtain the time constants of current deactivation at various repolarizing command voltages. When more than two exponential terms were needed to describe the current trajectories, the reported time constants are weighted averages of the best-fit time constants if necessary. To evaluate differences relative to WT, we generally applied a one-way ANOVA test, if data were homogeneous and normally distributed. Alternatively, we applied the non-parametric Kruskall–Wallis ANOVA test if data were not normally distributed. Additional statistical details are reported in the figure legends (main and supplemental) and Supplemental Table 2.

**Coarse grain simulations**. See Supplementary Methods (Supplementary Material, pp. 25–26).

**Atomistic simulations**. For computational expedience, to ease the translocation of the S4 Arginines[69], the CTC F256A mutant was introduced using PyMOL. The built structure was embedded in a POPC membrane using CHARMM-GUI[70] and solvated in 150 mM KCl solution. Protein, lipids and ions were modelled using the Charmm36M force field[71] and water was described using the TIP3P model. After energy minimization and equilibration, activating and deactivating simulations were performed by submitting the system to external electric fields yielding transmembrane potentials of +300 and −650 mV, respectively. Long-range electrostatic interactions were calculated using the particle mesh Ewald method[72] and hydrogen-bond lengths were constrained using LINCS[73]. Pressure and temperature were maintained through the use of the Parrinello–Rahman barostat (1 bar)[74] and v-rescale (300 K)[75] thermostat. The effects of S4 movement on T1-domain interactions were calculated using 750 ns of trajectory frames following the movement of R4 Arginine. Hydration profiles were calculated over 30 ns of MD simulations, using the Channel Annotation Package (CHAP)[76].

## Data availability
The cryo-EM maps of the human Kv3.1a structures and the corresponding atomic coordinates have been deposited in the Electron Microscopy Data Bank and the Protein Data Bank under the accession codes EMD-13416. /7PHH (apo), EMD-13417. /7PHI (zinc), EMD-13418/7PHK (dimer in zinc), EMD-13419/7PHL (EDTA), respectively. Source data for electrophysiological experiments (Fig. 6, Supplemental Figs. S11, S13, S14) and MD results (Fig. 7) are provided with this paper. Source data are provided with this paper.

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

## Acknowledgements

We thank Elizabeth Maclean, Dr. Loic Carrique and Dr. Helen Duyvesteyn at Oxford Particle Imaging Centre (Oxford, UK), and Dr. Christos Savva and Dr. T.J. Ragan at Midlands Regional Cryo-EM Facility (Leicester, UK) for their assistance with electron microscopes. We also thank Dr. Ashley C.W. Pike and Dr. Brian Marsden at Centre for Medicines Discovery (Oxford, UK) for their assistance with model building and cluster maintenance. This research was carried out with funding from the Innovative Medicines Initiative 2 Joint Undertaking (JU) under grants agreement No. 875510 (EUbOPEN) and No. 115766 (ULTRADD). The JU receives support from the European Union's Horizon 2020 research and innovation programme and EFPIA and Ontario Institute for Cancer Research, Royal Institution for the Advancement of Learning McGill University, Kungliga Tekniska Hoegskolan, Diamond Light Source Ltd. We also acknowledge support from a Wellcome strategic award (Grant No. 106169/Z/14/Z). Oxford Particle Imaging Centre was funded by a Wellcome Trust JIF award (Grant No. 060208/Z/00/Z) and is supported by equipment grants from WT (093305/Z/10/Z). M.C. received support from the Jefferson Synaptic Biology Center. A.S. is supported by a Marie Skłodowska-Curie grant 898762 (Lipopeutics). L.D. acknowledges SciLifeLab and the Swedish Research Council (VR 2018-04905) for funding. The MD simulations were performed on resources provided by the Swedish National Infrastructure for Computing (SNIC) on Beskow at the PDC Centre for High Performance Computing (PDC-HPC) and we acknowledge PRACE for awarding us access to Piz-Daint hosted at the Swiss national supercomputing center (CSCS).

## Author contributions

G.C., S.V., A.F.C., N.A.B.-B. and K.L.D. designed and cloned the constructs. The protein was expressed by G.C., N.K.S., G.M. and S.M.M.M., and the protein purification was done by G.C. and N.K.S. G.C., P.Q., P.C.-H., M.R., and K.S. collected the EM datasets. G.C. processed the EM data and built the model. A.S., J.B.C. and L.D. performed MD simulations. Q.L. and M.C. expressed the protein in oocytes and performed electrophysiology experiments. K.L.D., G.C., and A.S. wrote the manuscript. The manuscript was revised by K.L.D., G.C., A.S., Q.L., M.C., and L.D.

## Competing interests

The authors declare no competing interests.
