## [Peer Review File · Nature Communications]

REVIEWER COMMENTS

Reviewer #1 (Remarks to the Author):

This manuscript provides substantial new structural information about the factors that regulate the activity of Kv3.1 voltage-dependent potassium channels. These are the canonical rapidly activating/deactivating delayed rectifier channels that allow subsets of neurons to fire at high rates of hundreds of Hz. Among other aspects, the way the cytoplasmic N-terminal T1 domain interacts intimately with the core of the central membrane spanning regions including the S4/S5 linker and the S6T helix is novel and provides insights into these channels differ from other delayed rectifier K channels in their kinetic behavior and voltage-dependence. The clarification of the role of the "turret" region of Kv3 channels is also a useful contribution. I believe, however, that there are a number of places where the presentation could benefit from some clarifications.

1) The first sentence of the paragraph starting on line 144-162 is confusing and ambiguous. It states that "the structures of the isolated T1 domains for Kv4.2 and Kv4.3" show that their alpha6 conformations are more similar to those of Kv3.1 than those in "full-length structures from Kv family members." I'm assuming that the authors meant to say "... from Kv1 family members" but the referenced figures also show a Kv2 family channel, and, of course, these are all Kv family channels. It is therefore not clear if the authors are trying to say something about isolated vs. full-length structures.

2) The first amino acid substitution that is introduced and discussed in detail in the manuscript is the D81K mutation which MD simulations predict would alter the occupancy of K⁺ ions in the lower vestibule. It is speculated that D28 is required to help maintain the large conductance characteristics for Kv3 channels. Surprisingly, the effect of this mutation was not tested in the voltage-clamp experiments.

3) A MD simulation is carried out to examine the movement of the R1-R3 residues in the voltage sensor in response to brief depolarization (Fig. EV4, Movie M2). This is provided to help explain the effects of replacements of R332 in the S4/5 linker and K449 in S6T on voltage-dependence. It would therefore make sense to provide some evaluation of exactly how the upward movement would be affected by such replacements.

4) The functional role of the interactions between the T1 domain and the S4/S5 linker and the S6 tail were tested in voltage-clamp experiments in which these interactions were partially disrupted by introducing single amino acid substitutions. The results fall into two classes. The first class of mutations alters the voltage range over which the channel opens. For example, the K449A substitution in the S6T domain, which disrupts a salt bridge to residues in T1, shifts the G/V curve to more positive potentials. This is interpreted as the T1 interaction with K449 being required to stabilize the open state of the channel, and similar inferences are made for mutations with similar effect. While this is likely, however, one cannot be sure of this without knowing the effect of the mutations on the stability of the closed state, which could be independent of the open state. At least some discussion of this might be appropriate. In contrast, the second class of mutations that disrupt the T1-S4/S5 - S6 tail interactions produce little or no change in activation voltage but greatly slow activation/deactivation. This is clearly of prime relevance to the biological properties of Kv3 channels.

It would be helpful to have a clearer discussion of why there are two such very different effects of these mutations that disrupt the T1- S4/S5 - S6 tail interactions. The text gives the impression that, while some mutations produce both effects, they also can also be produced independently. Although there is a helpful discussion of how the interacting amino may be generally involved in gating, it would be useful to have more clarification of

possibilities for the two very different types of effects beyond the statement that the interactions have "an essential role for fast activation and deactivation".

Minor comment:

The abbreviations VSD and PD are used without ever being defined. While most investigators of ion channels will know this shorthand for voltage-sensing and pore domains, they should be spelled out explicitly. Similarly for PH.

Reviewer #2 (Remarks to the Author):

The manuscript by Chi et al. presents a cryo-EM structure of Kv3.1. This is an important channel from a physiological perspective and this work represents the first structural study of the channel. The structural work is done to a high standard and the structure will be of interest to channel physiologists and others interested in voltage-dependent cation channels. The structure of Kv3.1 is highly similar to that of Kv1.2, which was determined previously. Noteworthy differences with Kv1.2 are present, however, and these have functional and mechanistic implications that increase our understanding of Kv3.1 and Kv channels more broadly. For example, the structure of Kv3.1 includes a slightly shifted S4 helix. This observation supports the conventional model of voltage-sensor gating movements. The authors have also identified an interface between the cytosolic T1 domain and the transmembrane region in Kv3.1 that is absent in Kv1.2. They have shown that some of the residues in this interface change the gating properties of the channel when mutated, which suggests that this interface may contribute to the gating characteristics of Kv3.1 that distinguish it from some other Kv channels (including, for example fast gating kinetics). The manuscript is written clearly. Portions of the manuscript are written for those who have a good understanding of Kv structures. A more general presentation may appeal to a broader audience. The structure itself is interesting and advances our understanding of Kv channels. Some of the conclusions drawn from the structure seem a bit speculative or are slightly over-stated. Specific comments are below:

1) As reflected in some of the comments below, this reviewer thinks that the following conclusion in the abstract is over-stated: "we identify close communication between $\alpha 6$ helix of T1 domain, S4/S5 linker and S6T helix as responsible for the ultra-fast activation/deactivation and open state stabilisation that are unique to Kv3 channels." While mutations in the interface may make the kinetics slower, other regions are almost certainly involved as well. For example, as described in the introduction, there are other regions of the channel that have been identified (by other studies) that alter gating kinetics. This suggests that the kinetics of the process are more nuanced than implied by the phrase "responsible for the ultra-fast activation/deactivation" of the abstract.

2) The authors identify that the H336A mutation slows the activation kinetics. This is interesting. However, other residues in the interface between T1 and S1-S6 were mutated but their gating kinetics are not discussed. Current traces for the mutations are shown in Supp. Fig. S12. Can the activation time constants be determined from these traces? How do those results fit with the idea that the T1-TMD interface is important for fast gating kinetics?

3) Lines 122-143. ATM motif. Since the interaction of the ATM motif with T1 is a new feature and so much emphasis is placed on the interaction in the text, mutations that address the functional significance of this interaction (on channel properties) would be worth doing. Alternatively, some of the discussion about the ATM motif may be better

placed in the discussion section.

4) I find the following clause in the abstract to be a bit too strong: "These findings provide fundamentally new insights into gating control and disease mechanisms and guide strategies for the design of pharmaceutical drugs targeting Kv3 channels." I would remove the word "fundamentally" and add the word "may" before "guide".

5) Lines 82-83 of the introduction could be clarified. N-type inactivation is due to the T1 domain (ball-and-chain).

6) Line 115. Readers may be unfamiliar with the "Kv1.2-Kv2.1 chimera". Some clarification would help.

7) Line 155. MD simulation discussion. This paragraph seems a bit out of place – MD simulations have not been introduced, but this could be fixed with some rewording. It seems like a reasonable hypothesis that D81 residues help attract K+.

8) Line 166. "hydrophobic charge transfer center" is not defined or referenced.

9) Lines 173-175. It is mentioned that E213 "neutralizes the positive gating charges in S4". However, hydrogen bonds between E213 and gating arginine residues are not observed. I assume that the authors are speculating that the resting conformation of the voltage sensor might have interactions between E213 and the gating arginines. Since this is speculation, this should be clarified.

10) Lines 178-180: "It is conceivable that the more compact linkers could act like springs on both ends of S4 and hence speed up movement in either direction, hence accelerating the kinetics of VSD activation and deactivation, respectively." This speculation would be better placed in the discussion section. I suspect that the molecular bases of fast activation/deactivation may be more complicated/nuanced than this model.

11) Line 274- "Instead, D114 is sandwiched between M107 and M441 in S6T and introduces further negative repulsion with the negative partial charges on the methionine side chains. The D114A phenotype is thus consistent with a relative stabilization of the open state due to reduced electrostatic repulsion with adjacent residues. " Interaction with D114 is not clear from figures. This explanation seems somewhat over-speculative.

12) Figure 5B&D: dotted lines are drawn, suggesting hydrogen bonds between residues, but some distances are much too far for hydrogen bonds (some are greater than 4Å).

Reviewer #3 (Remarks to the Author):

The manuscript by Chi et al is a well-conducted study regarding structural characterization of human Kv3.1 channel. In this work, structures of both transmembrane domain and cytoplasmic domain are resolved, providing interesting mechanistic insight into the role of T1 domain in the characteristic gating behavior of Kv3.1 channel. These structural findings were further supported by MD simulations, electrophysiological recordings and mutation studies. Regarding to the MD simulations, authors employed both atomistic MD simulations

and coarse-grained simulations. Atomistic MD simulations confirmed the determined structures are in an open-conductive state and revealed the role of negatively charged residues in the cytoplasmic domain for attracting K⁺ ions that helps to maintain the large conductance characteristic for Kv3 channels. Coarse-grained simulations are used to identify lipid preference at two possible binding sites.

I have some questions regarding the analysis of MD data:

1. In the method part, authors described that one production run was performed for 5.5 microsecond, while the hydration profiles were calculated over 30 ns. Why did the authors only calculate the hydration profiles over 30ns of trajectory, while the simulations were carried out for much longer time scale?

2. The atomistic MD simulations were conducted with an external electric field at a transmembrane potential of 300mV. Did the authors observe any ion permeation event to support that the channel is in a conductive state?

3. During the 5.5 microsecond MD simulation, were the selectivity filter conformation remained in a conductive state as determined by cryo-EM?

REVIEWER COMMENTS

Reviewer #1 (Remarks to the Author):

This manuscript provides substantial new structural information about the factors that regulate the activity of Kv3.1 voltage-dependent potassium channels. These are the canonical rapidly activating/deactivating delayed rectifier channels that allow subsets of neurons to fire at high rates of hundreds of Hz. Among other aspects, the way the cytoplasmic N-terminal T1 domain interacts intimately with the core of the central membrane spanning regions including the S4/S5 linker and the S6T helix is novel and provides insights into these channels differ from other delayed rectifier K channels in their kinetic behavior and voltage-dependence. The clarification of the role of the "turret" region of Kv3 channels is also a useful contribution. I believe, however, that there are a number of places where the presentation could benefit from some clarifications.

1) The first sentence of the paragraph starting on line 144-162 is confusing and ambiguous. It states that "the structures of the isolated T1 domains for Kv4.2 and Kv4.3" show that their $\alpha6$ conformations are more similar to those of Kv3.1 than those in "full-length structures from Kv family members." I'm assuming that the authors meant to say "... from Kv1 family members" but the referenced figures also show a Kv2 family channel, and, of course, these are all Kv family channels. It is therefore not clear if the authors are trying to say something about isolated vs. full-length structures.

We thank the reviewer for pointing out the ambiguity of the sentence. The statement we were trying to make was that the greater similarity between Kv3 and Kv4 T1 domain structure deduced from superpositions with the isolated domains of Kv4.2/Kv4.3 and similarity inferred from sequence alignments (former Figure S6 d) suggest that full-length Kv4 may also have a more structured T1-S1 linker region compared to the published Kv1.2 and Kv1.3 structures.

While the current work was under review, a series of full-length Kv4.2 structures in complex with auxiliary subunits and alone were published [1] and the coordinates released. We removed the former panel d with the superpositions of isolated T1 domains of Kv4.2 and Kv4.3 because the value is limited with the more informative full-length structural information for human Kv4.2 now available. Furthermore, we replaced panel c in Figure S6 by a superposition of our Kv3.1 structure with the full-length Kv4.2 structure (pdb: 7F0J) and full-length Kv1.2. It shows that compared to Kv1.2, the T1 domain of Kv4.2 is twisted with respect to the TM region as well, but to a lesser extent than Kv3.1. It also shows that $\alpha6$ of the Kv4.2 T1 domain is indeed more similar to Kv3.1, but unlike Kv3.1, it is located at a greater distance from S4/S5L. We also added panel d to Figure S6 which shows that S6T in Kv4.2 interacts with residues in T1 as well, but the interaction is different and involves residues not conserved between Kv3 and Kv4 channels.

Due to the availability of the Kv4.2 structure, we also added a new panel to former Figure EV2 (now Figure S12) to compare the S4 arrangement and to highlight the presence of the salt bridge between R6 backbone oxygen and Arginine R311 in the S4/S5 linker (corresponding to contact A in Figure 7a). These additional observations are discussed in the respective sections of the manuscript.

2) The first amino acid substitution that is introduced and discussed in detail in the manuscript is the D81K mutation which MD simulations predict would alter the occupancy of K⁺ ions in the lower vestibule. It is speculated that D28 is required to help maintain the large conductance characteristics for Kv3 channels. Surprisingly, the effect of this mutation was not tested in the voltage-clamp experiments.

We have added new data characterizing the D81K mutant in TEVC experiments (Figure S11, panels f-i) and describe the phenotype in the main text (page XY). We injected equal amounts of WT and D81K mRNAs into

separate groups of oocytes and tested them on the same day after the injections (within a window of 6-8 h). We found that D81K expresses significantly smaller currents at full activation (+ 50 mV, Figure S11, panel h), whilst the voltage-dependence of activation is only slightly depolarized (Figure S11i) and the time constants of deactivation slightly accelerated (Fig. S14). This result was replicated with two independent batches of oocytes. The current reduction caused by D81K is robust ($P < 0.00001$) and may reflect a lowered unitary conductance, as suggested by the results from the MD simulations (Figure S11 d, e). However, single channel recordings outside the scope of this manuscript would be required to confirm this interpretation and rule out that the decreased whole-cell D81K current is not the result of lower surface expression and/or decreased P_{omax} .

At any rate, we also asked whether D81K would be more sensitive to changes in driving force resulting from a relatively large K efflux during periods of high activity (e.g., fast repetitive spiking). Given the size and shape of the oocyte, this change would be difficult to detect. Nevertheless, when pulsing at 67 Hz, we observed a small cumulative inhibition of the currents at +50 mV (<15%) that is consistent with K depletion (Figure R1), and this effect trended toward a greater inhibition of the D81K ($n = 6-7$ oocytes). This would be consistent with the putative ability of D81 to concentrate K near the inner mouth of the pore, and thereby determine the Kv3.1 conductance. Future single channel studies focused on D81 would be needed to test this hypothesis further and solve the mechanism.

Figure R1. Cumulative inhibition of currents after repetitive pulsing to 50 mV at 67 Hz for wildtype Kv3.1a (black trace) and D81K mutant (red trace). Data are mean \pm SD from 6 and 5 oocytes, respectively.

3) A MD simulation is carried out to examine the movement of the R1-R3 residues in the voltage sensor in response to brief depolarization (Fig. EV4, Movie M2). This is provided to help explain the effects of replacements of R332 in the S4/5 linker and K449 in S6T on voltage-dependence. It would therefore make sense to provide some evaluation of exactly how the upward movement would be affected by such replacements.

To confirm the robustness of our prediction, we carried out new simulations, this time in the presence of a hyperpolarizing field. We ran two independent simulations under these conditions (Figure S15 b-c), and verified the correlation between voltage-sensor activation and the formation of the interaction between E116 in the S4/5 linker and K449 in S6T. We have now added two new panels to the former Figure EV4 (now: Figure 7 d-e) to show this correlation. This makes it rather unnecessary to probe the effect of mutations in silico. Indeed, rather than performing mutagenesis, it is possible to probe the propensity to form and break different types of interactions directly from MD simulations, for instance by measuring distance along time, as we show here.

4) The functional role of the interactions between the T1 domain and the S4/S5 linker and the S6 tail were tested in voltage-clamp experiments in which these interactions were partially disrupted by introducing single amino acid substitutions. The results fall into two classes. The first class of mutations alters the voltage range over which the channel opens. For example, the K449A substitution in the S6T domain, which disrupts a salt bridge to residues in T1, shifts the G/V curve to more positive potentials. This is interpreted as the T1 interaction with K449 being required to stabilize the open state of the channel, and similar inferences are made for mutations with similar effect. While this is likely, however, one cannot be sure of this without being knowing the effect of the mutations on the stability of the closed state, which could be independent of the open state. At least some discussion of this might be appropriate.

The reviewer is correct. Without additional experiments (e.g., voltage-dependence of gating currents and kinetic analysis of unitary currents) we cannot conclusively determine whether we have stabilization of the open state or destabilization of the closed state and vice versa. That is why GV shifts are only described as **relative** stabilization/destabilization of the open/closed state. It is, however, important to point out that the shift induced by K449A (and other mutations) is associated with acceleration of deactivation and slowing of activation. Because of this, we think that some mutations are having effects on the stability of activated and deactivated states of the voltage sensors, rather than only affecting the opening and closing of the pore. That is, mutations, directly or indirectly, mainly affect the depth of the voltage sensor's energy wells for the activated (up) and deactivated (down) states. As requested by the reviewer, we clarify/discuss these possibilities. There is also a bit of problem with terminology because -for the above reasons- activation/deactivation do not necessarily include opening and closing, as it is established by canonical models of voltage-dependent gating.

As the reviewer pointed out, there are two types of effects: 1) mutations that shift the GV curve and produce the corresponding expected effects (see above) on time constants of deactivation and activation; and 2) mutations that do not shift the GV curve but induce nearly equivalent changes in activation and deactivation, which then suggests that, rather than affecting the depth of the wells, these mutations change the height of the energy barrier of voltage sensor activation. In the discussion, we also point out that the phenotypes of some of the mutants could be due to impaired coupling between VSD and pore.

In contrast, the second class of mutations that disrupt the T1- S4/S5 - S6 tail interactions produce little or no change in activation voltage but greatly slow activation/deactivation. This is clearly of prime relevance to the biological properties of Kv3 channels.

It would be helpful to have a clearer discussion of why there are two such very different effects of these mutations that disrupt the T1- S4/S5 - S6 tail interactions. The text gives the impression that, while some mutations produce both effects, they also can also be produced independently. Although there is a helpful discussion of how the interacting amino may be generally involved in gating, it would be useful to have more clarification of possibilities for the two very different types of effects beyond the statement that the interactions have "an essential role for fast activation and deactivation".

We thank the reviewers for their thoughtful comment, and we have made changes to the manuscript to further clarify the functional differences we observe for the investigated mutants. For instance, we added the voltage-dependence of activation and deactivation time constants to the Supplementary Material (Figure S14) and discuss the respective phenotypes in the result section. The newly included data highlights that mutant H336 shows a significant reduction in both activation and deactivation kinetics (and hence shows no substantial change in $V_{0.5}$ compared to WT). In contrast, R339A shows faster activation and deactivation without changing $V_{0.5}$. These S4/S5 linker mutants are behaving therefore quite different from the mutants that cause a strong change in $V_{0.5}$ which can be explained by consistent effects on both activation and deactivation, eg mutants R332 and K449 exhibit faster deactivation and slower activation time constants (Fig S14 c,f). D114A shows a reversed phenotype, ie. faster activation and slower deactivation kinetics (Figure S14A).

The reviewer correctly states that in the absence of a structure describing Kv3.1 in a closed, deactivated conformation it is speculative to conclude that the salt bridge we observe between K449 in S6 and T1 residue E116 selectively stabilizes the open state. Future work presenting additional structures in other states, as well as measurements of gating currents to compare the effects on G/V versus Q/V curves for the described mutants would be needed to shed further light on the functional relevance of this salt-bridge on Kv3.1 gating.

Minor comment:

The abbreviations VSD and PD are used without ever being defined. While most investigators of ion channels will know this shorthand for voltage-sensing and pore domains, they should be spelled out explicitly. Similarly for PH.

Thanks for the comment. We have defined the abbreviations VSD, PD and PH in the introduction (page 3, paragraph 2 of the manuscript).

Reviewer #2 (Remarks to the Author):

The manuscript by Chi et al. presents a cryo-EM structure of Kv3.1. This is an important channel from a physiological perspective and this work represents the first structural study of the channel. The structural work is done to a high standard and the structure will be of interest to channel physiologists and others interested in voltage-dependent cation channels. The structure of Kv3.1 is highly similar to that of Kv1.2, which was determined previously. Noteworthy differences with Kv1.2 are present, however, and these have functional and mechanistic implications that increase our understanding of Kv3.1 and Kv channels more broadly. For example, the structure of Kv3.1 includes a slightly shifted S4 helix. This observation supports the conventional model of voltage-sensor gating movements. The authors have also identified an interface between the cytosolic T1 domain and the transmembrane region in Kv3.1 that is absent in Kv1.2. They have shown that some of the residues in this interface change the gating properties of the channel when mutated, which suggests that this interface may contribute to the gating characteristics of Kv3.1 that distinguish it from some other Kv channels (including, for example fast gating kinetics). The manuscript is written clearly. Portions of the manuscript are written for those who have a good understanding of Kv structures. A more general presentation may appeal to a broader audience. The structure itself is interesting and advances our understanding of Kv channels. Some of the conclusions drawn from the structure seem a bit speculative or are slightly over-stated. Specific comments are below:

1) As reflected in some of the comments below, this reviewer thinks that the following conclusion in the abstract is over-stated: “we identify close communication between $\alpha 6$ helix of T1 domain, S4/S5 linker and S6T helix as responsible for the ultra-fast activation/deactivation and open state stabilisation that are unique to Kv3 channels.” While mutations in the interface may make the kinetics slower, other regions are almost certainly involved as well. For example, as described in the introduction, there are other regions of the channel that have been identified (by other studies) that alter gating kinetics. This suggests that the kinetics of the process are more nuanced than implied by the phrase “responsible for the ultra-fast activation/deactivation” of the abstract.

Following the advice of this reviewer, we changed the sentence in the abstract to “we identify several residues in the S4/S5 linker which influence the gating kinetics and an electrostatic interaction between acidic residues in $\alpha 6$ of T1 and R449 in the pore-flanking S6T helices”.

2) The authors identify that the H336A mutation slows the activation kinetics. This is interesting. However, other residues in the interface between T1 and S1-S6 were mutated but their gating kinetics are not discussed. Current traces for the mutations are shown in Supp. Fig. S12. Can the activation time constants be determined from these traces? How do those results fit with the idea that the T1-TMD interface is important for fast gating kinetics?

In the revised version of our manuscript, we have now included activation time constants for all the investigated mutants in the T1/S4/S5 linker interface and for S6T mutant K449A (Supplementary Figure S14). The data highlights that for most of the other mutants with changes in $V_{0.5}$, the voltage-dependence of time

constants is shifted accordingly, resulting in faster activation and slower deactivation kinetics (eg. for R332A). Only mutants H336A and R339A in the S4/S5 linker result in slower or faster kinetics across the whole voltage range and hence leave $V_{0.5}$ largely unaffected. A possible mechanistic interpretation of these phenotypes is described in the response to reviewer comments above.

3) Lines 122-143. ATM motif. Since the interaction of the ATM motif with T1 is a new feature and so much emphasis is placed on the interaction in the text, mutations that address the functional significance of this interaction (on channel properties) would be worth doing. Alternatively, some of the discussion about the ATM motif may be better placed in the discussion section.

We moved the section about the potential functional importance of the ATM motif and disease variants affecting this region of the channel to the discussion as suggested, because analysis of further mutants in the ATM motif is beyond the scope of the current work.

4) I find the following clause in the abstract to be a bit too strong: “These findings provide fundamentally new insights into gating control and disease mechanisms and guide strategies for the design of pharmaceutical drugs targeting Kv3 channels.” I would remove the word “fundamentally” and add the word “may” before “guide”.

The sentence in the abstract has been modified as suggested by the reviewer.

5) Lines 82-83 of the introduction could be clarified. N-type inactivation is due to the T1 domain (ball-and-chain).

We are grateful that the reviewer spotted this misleading statement in the introduction. We have changed this section as follows and a reference was added for the ball-and-chain N-type inactivation:

“For the Kv1 subfamily of potassium channels, characterization of T1-deleted Shaker variants [22] has ruled out an essential role of the T1 domain for normal gating beyond the well-studied role of the N-terminus for N-type inactivation [23].”

6) Line 115. Readers may be unfamiliar with the “Kv1.2-Kv2.1 chimera”. Some clarification would help.

A sentence has been added to clarify the expression “Kv1.2-Kv2.1 chimera”:

“...similar to the structures of rat Kv1.2 [18] and the Kv1.2-Kv2.1 chimera, in which the voltage-sensing S3b/S4 segment from Kv2.1 has been grafted onto Kv1.2 for structure determination”

7) Line 155. MD simulation discussion. This paragraph seems a bit out of place – MD simulations have not been introduced, but this could be fixed with some rewording. It seems like a reasonable hypothesis that D81 residues help attract K^+ .

We agree that the discussion of the MD results for the D81K mutant was missing context and we therefore reworded this section as suggested by the reviewer. We also discuss new functional data showing the reduced ionic conductance of mutant D81K (panels f-j added to Supplementary Figure S11) which may further support our hypothesis that D81 may help to attract K^+ ions. See response to reviewer 2 also.

8) Line 166. “hydrophobic charge transfer center” is not defined or referenced.

The term has been referenced to Tao *et al.* 2010 [2].

9) Lines 173-175. It is mentioned that E213 “neutralizes the positive gating charges in S4”. However, hydrogen bonds between E213 and gating arginine residues are not observed. I assume that the authors are speculating that the resting conformation of the voltage sensor might have interactions between E213 and the gating arginines. Since this is speculation, this should be clarified.

We thank the reviewer for bringing our attention to this. We corrected the sentence to reflect the fact that we indeed do not observe hydrogen bonds between these two residues in our structure and we instead only suspect that this interaction might be possible in a different VSD conformation where S4 is shifted to bring the two residues closer to each other. This has been corrected accordingly, as follows:

“Another notable difference is that a second glutamate, E213 in the S2 segment of Kv3.1a, replaces T184 in the Kv1.2 structure (Figure S12 a, b). It may therefore neutralize positive gating charges in S4 when the VSD assumes a different activation state than captured in the current structure, in which the residues are too far apart for this putative interaction.”

10) Lines 178-180: “It is conceivable that the more compact linkers could act like springs on both ends of S4 and hence speed up movement in either direction, hence accelerating the kinetics of VSD activation and deactivation, respectively.” This speculation would be better placed in the discussion section. I suspect that the molecular bases of fast activation/deactivation may be more complicated/nuanced than this model.

As suggested, we removed this sentence from the result section because it is speculative.

11) Line 274- “Instead, D114 is sandwiched between M107 and M441 in S6T and introduces further negative repulsion with the negative partial charges on the methionine side chains. The D114A phenotype is thus consistent with a relative stabilization of the open state due to reduced electrostatic repulsion with adjacent residues. “ Interaction with D114 is not clear from figures. This explanation seems somewhat over-speculative.

We have removed the speculative explanation for the D114 phenotype and focused our interpretation on other residues for which interactions can be clearly seen in the structure. We also included voltage-dependent activation and deactivation time constants for this mutant to Supplementary Figure S14A, which shows that the mutation results in faster activation kinetics and slower deactivation kinetics compared to wildtype Kv3.1a, which correlates well with the mutant’s hyperpolarizing shift in $V_{0.5}$ determined from the G/V curve.

12) Figure 5B&D: dotted lines are drawn, suggesting hydrogen bonds between residues, but some distances are much too far for hydrogen bonds (some are greater than 4Å).

This is a valid point raised by the reviewer. We have changed the Figure panels as requested and the revised version of Figure 5 (renamed to Figure 8) only shows dotted lines or inter-atom distances which would allow the formation of hydrogen bonds (4 Å or smaller).

Reviewer #3 (Remarks to the Author):

The manuscript by Chi et al is a well-conducted study regarding structural characterization of human Kv3.1 channel. In this work, structures of both transmembrane domain and cytoplasmic domain are resolved,

providing interesting mechanistic insight into the role of T1 domain in the characteristic gating behavior of Kv3.1 channel. These structural findings were further supported by MD simulations, electrophysiological recordings and mutation studies. Regarding to the MD simulations, authors employed both atomistic MD simulations and coarse-grained simulations. Atomistic MD simulations confirmed the determined structures are in an open-conductive state and revealed the role of negatively charged residues in the cytoplasmic domain for attracting K⁺ ions that helps to maintain the large conductance characteristic for Kv3 channels. Coarse-grained simulations are used to identify lipid preference at two possible binding sites.

I have some questions regarding the analysis of MD data:

1. In the method part, authors described that one production run was performed for 5.5 microsecond, while the hydration profiles were calculated over 30 ns. Why did the authors only calculate the hydration profiles over 30ns of trajectory, while the simulations were carried out for much longer time scale?

The hydration profiles from MD simulations were calculated with the aim of validating that the pore was open in the resolved structure. Thus, only the initial 30 ns of the trajectory was used to 'annotate' the structure using the CHAP package.

The simulations were indeed performed under a transmembrane potential, resulting in a deviation of the protein structure away from this cryo-EM resolved configuration. Thus, calculating hydration profiles over the entire 5.5 microsecond would no longer be an annotation of the resolved structure but would be an analysis of configurations sampled by MD simulations.

2. The atomistic MD simulations were conducted with an external electric field at a transmembrane potential of 300 mV. Did the authors observe any ion permeation event to support that the channel is in a conductive state?

Consistent with the observation of the pore being in a dilated conformation with a radius enough to accommodate a hydrated K⁺ ion, we observed multiple ions entering the central cavity from the intracellular solution. The ion permeation pathway primarily involved ions passing through the fenestrations between the T1 and TMD domains – and reaching the central pore near the D81 residues, and finally up through the PVP hinge region. We provide a movie (file Movie_R3.mp4) showing various potassium ions entering the region below the selectivity filter (beyond which ion movement is blocked by SF collapse – see answer to Q3 below).

3. During the 5.5 microsecond MD simulation, were the selectivity filter conformation remained in a conductive state as determined by cryo-EM?

Similar to our prior work [3] with other voltage-gated ion channels, we observe selectivity-filter collapse after the exit of the resolved ions. This has primarily been attributed to the inability of classical MD force fields to describe polarizability within this region. Here, similar to prior work [3, 4], the collapse was primarily caused by the backbone of Y403 residue (see plots in Figure R2 where the residue's dihedral angle changes after ~200 ns).

Figure R2. Change in dihedral angles of residue Y403 in the selectivity filter of Kv3.1a for chains A-D during the course of 1 μsec.

References

1. Kise, Y., et al., *Structural basis of gating modulation of Kv4 channel complexes*. Nature, 2021. **599**(7883): p. 158-164.
2. Tao, X., et al., *A gating charge transfer center in voltage sensors*. Science, 2010. **328**(5974): p. 67-73.
3. Kasimova, M.A., et al., *Helix breaking transition in the S4 of HCN channel is critical for hyperpolarization-dependent gating*. Elife, 2019. **8**.
4. Shrivastava, I.H. and M.S. Sansom, *Simulations of ion permeation through a potassium channel: molecular dynamics of KcsA in a phospholipid bilayer*. Biophys J, 2000. **78**(2): p. 557-70.

REVIEWERS' COMMENTS

Reviewer #1 (Remarks to the Author):

The authors have responded to each of my comments with thoughtful changes and/or new data. They should be congratulated on their work.

Reviewer #2 (Remarks to the Author):

The authors have addressed my major comments. Below are some minor comments that may improve the manuscript.

References needed for lines 40-46.

Line 59 (and in other portions of the text): references are made to the "PVP hinge region". My understanding is that the PVP motif, while it forms a bend, may not act as a "hinge". A glycine residue primarily acts as a hinge for gating of the pore. The "PVP motif" would be more accurate. A reference would be helpful for the function of the PVP motif in line 59.

line 86-87. claims such as "first" may not be allowed.

Lines 164-165: Wording is a bit confusing. Four arginines (R1-R4) are primarily responsible for voltage sensing (not six). In Kv1.2-2.1, these positions are above the CTC. In Kv3.1, it seems that R1-R3 above and one (R4) is below.

Line 356-358: Q/V and G/V are terms that will not be common to the general reader.

S. Table 1. Ramachandran Favoured and Disallowed statistics seem to be reversed.

fig S2. Panel A, blue structures seem to have different handedness. It would be best to show the correct hand. Check all such panels in the supplementary figures.

fig S6. Panel A. the surface is not defined. Is it a molecular surface?

fig S7. What do dotted lines represent? Cys 472 is the final ordered residue. It seems unlikely that H-bonds would be well defined by the density. The interaction between C472 and W36 does not appear to have the correct geometry for a H-bond.

fig S12. Why is pdb 3LUT compared? Wouldn't a higher-resolution structure of Kv1.2-2.1 be better to compare (2R9R)?

Reviewer #3 (Remarks to the Author):

All of my previous questions are properly addressed.

Response to reviewer's comments for NCOMMS-21-34839A

We thank all reviewers for reading through a second time and appreciate their attention to detail. The remaining issues raised by reviewer #2 have been addressed now, as outlined below.

Reviewer #1 (Remarks to the Author):

The authors have responded to each of my comments with thoughtful changes and/or new data. They should be congratulated on their work.

Reviewer #2 (Remarks to the Author):

The authors have addressed my major comments. Below are some minor comments that may improve the manuscript.

References needed for lines 40-46.

We thank the reviewer for the thoughtful comment and have added references for this section about the physiological relevance of Kv3 channels.

Line 59 (and in other portions of the text): references are made to the "PVP hinge region". My understanding is that the PVP motif, while it forms a bend, may not act as a "hinge". A glycine residue primarily acts as a hinge for gating of the pore. The "PVP motif" would be more accurate. A reference would be helpful for the function of the PVP motif in line 59.

Following the reviewer's suggestion, we replaced the term "PVP hinge" with PVP motif or PVP region throughout text and figure legends. We also added a sentence with references to several studies investigating the potential role of the PXP motif for gating of Kv channels.

line 86-87. claims such as "first" may not be allowed.

We removed the word "first" from the sentence.

Lines 164-165: Wording is a bit confusing. Four arginines (R1-R4) are primarily responsible for voltage sensing (not six). In Kv1.2-2.1, these positions are above the CTC. In Kv3.1, it seems that R1-R3 above and one (R4) is below.

We changed the phrasing to make it more clear that mainly R1-R4 in the arginine-rich motif in S4 of voltage-gated channels are voltage-sensing. This is also reflected by a different color coding in Figure 3, showing only R1-R4 in blue. The Figure legend of Figure 3 has also been corrected accordingly.

Line 356-358: Q/V and G/V are terms that will not be common to the general reader.

Thanks for raising this. We have rephrased the sentence to call these gating charge-voltage (Q-V) and conductance-voltage (G/V) relationship, respectively. The reference 57 cited here is a review explaining the term for the non-expert reader.

S. Table 1. Ramachandran Favoured and Disallowed statistics seem to be reversed.

This mistake in the previous version of Suppl. Table 1 has been corrected.

fig S2. Panel A, blue structures seem to have different handedness. It would be best to show the correct hand. Check all such panels in the supplementary figures.

The handedness has been inverted to the correct one used for model building.

fig S6. Panel A. the surface is not defined. Is it a molecular surface?

The figure legend of Fig. S6 A has been updated to define the shown surface which represents Kv1.3/ β 2.1

fig S7. What do dotted lines represent? Cys 472 is the final ordered residue. It seems unlikely that H-bonds would be well defined by the density. The interaction between C472 and W36 does not appear to have the correct geometry for a H-bond.

We corrected the Figure and removed the dotted lines between C472 and W36.

fig S12. Why is pdb 3LUT compared? Wouldn't a higher-resolution structure of Kv1.2-2.1 be better to compare (2R9R)?

Following the suggestion of the reviewer, panels b and e of Figure S12 have been redrawn with pdb 2R9R instead of pdb 3LUT and the labels/figure legend were updated accordingly.

Reviewer #3 (Remarks to the Author):

All of my previous questions are properly addressed.